Planktonic associations between medusae (classes Scyphozoa and Hydrozoa) and epifaunal crustaceans

Muffett Kaden kmmuffett@tamu.edu
Miglietta Maria Pia
Department of Marine Biology, Texas A&M University - Galveston , Galveston, Texas , United States
Dos Santos Antonina
Electronic publication date: 2021 Apr 23
Publication date: 2021
Volume: 9
Electronic Location ID: e11281
Received 2020 Dec 17; Accepted 2021 Mar 24
Copyright: © 2021 Muffett and Miglietta
Copyright year: 2021
Copyright holder: Muffett and Miglietta
License: This is an open access article distributed under the terms of the Creative Commons Attribution License, which permits unrestricted use, distribution, reproduction and adaptation in any medium and for any purpose provided that it is properly attributed. For attribution, the original author(s), title, publication source (PeerJ) and either DOI or URL of the article must be cited.
License URL: https://creativecommons.org/licenses/by/4.0/

Keywords: Hydrozoa, Scyphozoa, Crustacea, Association, Commensal, Epifauna, Marine, Jellyfish, Medusa

Funding: The authors received no funding for this work.

==============================
Jellyfish are known to carry various epibionts, including many of the subphylum Crustacea. However, the associations between gelatinous zooplankton and other invertebrates have been chronically overlooked. Crustacea, a massive clade of economically, ecologically, and culturally important species, includes many taxa that utilize gelatinous zooplankton for food, transport, and protection as both adults and juveniles. Here we compile 211 instances of epifaunal crustaceans recorded on Hydromedusae and Scyphomedusae from a century of literature. These include 78 identified crustacean species in 65 genera across nine orders found upon 37 Hydromedusa species and 48 Scyphomedusae. The crustacean life stage, location, nature of the association with the medusa, years, months, and depths are compiled to form a comprehensive view of the current state of the literature. Additionally, this review highlights areas where the current literature is lacking, particularly noting our poor understanding of the relationships between juvenile crabs of commercially valuable species and medusae.

Background

An increased focus on ocean climate research in the past 20 years has made clear the fragility of the world’s oceans and the organisms that live within them. The rate at which species are disappearing, undergoing climate-related range fluctuations, and experiencing developmental and behavioral changes is unlike anything seen in the time of record (Walther et al., 2002; Guinotte & Fabry, 2008; Comeaux, Allison & Bianchi, 2012). Attempts to model changes in populations, species, and ecosystems have laid bare the degree to which dynamics among many marine invertebrates remain unknown and poorly understood (Uye, 2008; Brodeur, Ruzicka & Steele, 2011; Henschke et al., 2014). This problem is especially apparent in jellyfish of the phylum Cnidaria, which are chronically understudied and poorly categorized (Riascos et al., 2013; Gambill & Peck, 2014; Sweetman et al., 2016; Gómez Daglio & Dawson, 2017). Long considered a pure pest, the last decade has demonstrated an increasing number of ways in which jellyfish are critical components of the ecosystems they reside in (Cardona et al., 2012; Hays, Doyle & Houghton, 2018). While they are best known for the vertebrates that depend on them for nutrition, including turtles and birds, they provide a host of ecosystem services unrelated to a “prey” designation. Reef and non-reef fish juveniles readily congregate around large scyphozoans, some hiding within the bell or between tentacles when disturbed (Brodeur, 1998; D’Ambra et al., 2014; Tilves et al., 2018). Large jellyfish can reach sizes that allow them to support independent encrusting organisms, like barnacles and brittle stars (Ohtsuka et al., 2010; Álvarez-Tello, López-Martínez & Rodríguez-Romero, 2013; Yusa et al., 2015).

While research has expanded around services jellyfish provide (Riascos et al., 2018), much of this research focuses on benefit and harm to vertebrates (Brodeur, 1998; Cardona et al., 2012; Mir-Arguimbau, Sabatés & Tilves, 2019). However, the relationships between scyphomedusae, hydromedusae and other invertebrates are currently poorly characterized. A prime invertebrate group to analyze through this lens is Crustacea. Crustaceans are some of the most visible and well-studied marine invertebrates. They are present in every region and are integral components of food webs, including species of high commercial value and known ecological significance (Boudreau & Worm, 2012). Ecological processes that impact them are thus relevant to humans. However, studies focusing on epifaunal crustaceans and jellyfish interactions have been scarce, incomplete, and taxonomically imprecise. Moreover, such studies are often narrowly focused accounts of interactions with single individuals (Weymouth, 1910; Reddiah, 1968; Yusa et al., 2015). Some early communications discuss these interactions as common knowledge that has, however, failed to be recorded in the scientific literature (Jachowski, 1963). This review provides a list of documented crustacean epibionts on medusae of the orders Scyphozoa and Hydrozoa. This work aims to assess the breadth and depth of jellyfish-crustacean interaction and develop a resource for further studies.

Methodology

Four independent sets of searches were conducted in Google Scholar using keywords, as described in Fig. 1. All four searches were conducted in early November 2019 and were revisited in January 2021 to include all results through the end of 2019. Searches were performed in English, and as such, only papers published in or with an available translation to English were included. The number of papers yielded by each of the four searches is shown in Fig. 1, ranges from 4,840 articles (for keywords Crustacea, Scyphozoa) to 13,300 (for keywords Crustacea, Jellyfish) (See Fig. 1 for details). Only papers in which the primary focus was associations between medusae (Hydrozoa and Scyphozoa) and crustaceans were further selected.

Figure 1 Summary of Google Search Results.

The number of results reported by Google Scholar Advanced Search where both “Crustacea” and one of the four medusa describer terms was included (“Hydrozoa”, “Scyphozoa”, “medusa”, or “jellyfish”) and at least one of the following terms was included (Association, Associated, Symbiotic, Symbiosis, Commensal, Epifaunal, Harboring, Parasitic, Parasitoid, Epibiont or Epibiotic).

The four searches performed returned many invariable results. All titles and abstracts were checked for relevance. Results from 161 papers were obtained initially and then narrowed to 81, after excluding repeat papers mistakenly included multiple times and papers on cubomedusae, ctenophores, ascidians, and non-crustacean epibionts. Also, results from six relevant literature reviews were included (Vader, 1972; Pagès, 2000; Towanda & Thuesen, 2006; Ohtsuka et al., 2011; Schiariti et al., 2012; Wakabayashi, Tanaka & Phillips, 2019). These reviews account for 40 interactions from 29 sources (Table 1). The inclusion of the literature reviews was deemed essential to include results from earlier sources and non-English sources not available on Google Scholar. Results from literature reviews that had no information on the nature of the interaction between the medusa and crustaceans (such as taxa identification, location, etc.) were eliminated. Records were also analyzed for taxon validity using the World Register of Marine Species (WoRMS). Seven papers within the database that referred to invalid taxa with no valid synonymized name in WoRMS were removed. Results from 97 unique sources (68 articles from the Google Scholar search and 29 from literature reviews) were kept. From these 97 sources, 211 distinct interactions were extracted. Details provided by each paper were recorded in Table 1.

Table 1 Associations reported organized by host.

Every association in all reviewed papers with details on species and higher order classification of host, species of associate, sex and life stage of associate, notes on association, location on host, location association was recorded, date of record, depth of association and literature source.

Host Species	Epibiont	Notes	Life Stage and Sex	Location on Medusa	Location	Collection	Limited	Month/Year	Depth	Reference	
Scyphozoa											
Coronatae											
Nausithoe rubra Vanhöffen, 1902	Prohyperia shihi Gasca, 2005	Not visibly parasitizing host, female and male pair	F, M	EX	Gulf of California	ROV	L	2012 Feb	907 m	Gasca, 2013	
Rhizostomeae											
Acromitoides purpurus Mayer, 1910	Charybdis feriata Linnaeus, 1758	Never more than one per medusa	?	?	Various bays, Philippines	HC	N	2014–2015, Feb–Apr	NS	Boco & Metillo, 2018	
Acromitoides purpurus Mayer, 1910	Paramacrochiron sp.	Present 44–100% of medusae depending on location and medusa color morph	?	?	Various bays, Philippines	HC	N	2014–2015, Feb–Apr	NS	Boco & Metillo, 2018	
Acromitus flagellatus Maas, 1903	Latreutes anoplonyx Kemp, 1914	N/A	?	?	Indonesia	?	?	?	?	Hayashi, Sakagami & Toyoda, 2004	
Acromitus sp.	Hourstonius pusilla K.H. Barnard, 1916	Present throughout the adult medusa population	?	SUM, O	Chilka Lake, India	?	L	?	?	Chilton, 1921 via Vader, 1972	
Cassiopea sp.	Ancylomenes aqabai Bruce, 2008	N/A	OF & F	O	Aqaba, Jordan	HC	L	1976 Mar	NS	Bruce, 2008	
Cassiopea sp.	Ancylomenes holthuisi Bruce, 1969	N/A	?	O	Zanzibar harbour	SC	L	1970 Dec	20-25 m	Bruce, 1972	
Cassiopea sp.	Periclimenes pedersoni Chace, 1958	N/A	OF & M	O	Santa Marta, Colombia	?	N	?	3-40 m	Criales, 1984	
Cassiopea sp.	Periclimenes tonga Bruce, 1988	N/A	OF	?	Nuapapu Island (southside), Vava’u Group, Tonga	?	L	1985 Jul	?	Bruce, 1988	
Cassiopea sp.	Periclimenes yucatanicus Ives, 1891	N/A	OF & jM & F	O	Santa Marta, Colombia	?	N	?	3–25 m	Criales, 1984	
Cassiopea sp.	Sewellochiron fidens Humes, 1969	N/A	F, M	?	Puerto Rico	?	?	1959	3 m	Humes, 1969	
Catostylus mosaicus Quoy & Gaimard, 1824	Acartia sp.	N/A	C & A	O	Botany Bay, Lake Illawarra, Smiths Lake, New South Wales	HC	N	1999–2000	NS	Browne & Kingsford, 2005	
Catostylus mosaicus Quoy & Gaimard, 1824	Cymodoce gaimardii H. Milne Edwards, 1840	Autumnal prevalence peak	?	O, SUM, EX	Port Phillip Bay,Victoria	HC	N	2009 Aug–2010 Sep	NS	Browne, 2015	
Catostylus mosaicus Quoy & Gaimard, 1824	Cymodoce gaimardii H. Milne Edwards, 1840	Highest prevalence in Mar	A & J	B, O	Port Phillip Bay,Victoria	HC	N	2008 Aug– 2010 Sep	NS	Browne, Pitt & Norman, 2017	
Catostylus mosaicus Quoy & Gaimard, 1824	Evadne sp.	Only one specimen	?	O	Botany Bay, New South Wales	HC	L	1999-2000	NS	Browne & Kingsford, 2005	
Catostylus mosaicus Quoy & Gaimard, 1824	Hyperia gaudichaudii H. Milne Edwards, 1840	September prevalence peak, Es and Js embedded in host tissue	E & J & A	GVC, B	Port Phillip Bay,Victoria	HC	N	2008 Aug– 2010 Sep	NS	Browne, 2015	
Catostylus mosaicus Quoy & Gaimard, 1824	Ibacus sp.	A single specimen from Sydney museum collection	PL	SUB	Hawkesbury River, New South Wales	?	L	1925	?	Thomas, 1963	
Catostylus mosaicus Quoy & Gaimard, 1824	Latreutes anoplonyx Kemp, 1914	Found on medusa type specimen from Pakistan	OF & J	O	Korangi Creek, Pakistan	HC	L	1995	NS	Tahera & Kazmi, 2006	
Catostylus mosaicus Quoy & Gaimard, 1824	Lucifer sp.	N/A	?	O	Botany Bay, Lake Illawarra, New South Wales	HC	L	1999–2000	NS	Browne & Kingsford, 2005	
Catostylus mosaicus Quoy & Gaimard, 1824	Oithona sp.	Only present on two medusae in one lake	?	O	Lake Illawarra, New South Wales	HC	L	1999–2000	NS	Browne & Kingsford, 2005	
Catostylus mosaicus Quoy & Gaimard, 1824	Oncaea sp.	N/A	?	O	Botany Bay, Smiths Lake, New South Wales	HC	L	1999–2000	NS	Browne & Kingsford, 2005	
Catostylus mosaicus Quoy & Gaimard, 1824	Oncaea venusta Philippi, 1843	N/A	?	O	Botany Bay, Lake Illawarra, New South Wales	HC	L	1999–2000	NS	Browne & Kingsford, 2005	
Catostylus mosaicus Quoy & Gaimard, 1824	Paramacrochiron maximum Thompson I.C. & Scott A., 1903	Present in hundreds per medusa at all phases of development and size class	A & J & OF	O	Botany Bay, Lake Illawarra, New South Wales	HC	N	1999–2000	NS	Browne & Kingsford, 2005	
Catostylus mosaicus Quoy & Gaimard, 1824	Pseudodiaptomus sp.	N/A	A	O	Botany Bay, Lake Illawarra, New South Wales	HC	N	1999–2000	NS	Browne & Kingsford, 2005	
Catostylus mosaicus Quoy & Gaimard, 1824	Temora sp.	N/A	A	O	Botany Bay, Lake Illawarra, Smiths Lake, New South Wales	HC	N	1999–2000	NS	Browne & Kingsford, 2005	
Catostylus mosaicus Quoy & Gaimard, 1824	Tortanus barbatus Brady, 1883	N/A	C & A	O	Botany Bay, Lake Illawarra, New South Wales	HC	N	1999–2000	NS	Browne & Kingsford, 2005	
Catostylus sp.	Charybdis feriata Linnaeus, 1758	Present from Apr–May	?	O, SUM	Kolambugan, Lanao del Norte	?	N	2013 Dec– 2014 Jul	NS	Boco, Metillo & Papa, 2014	
Catostylus sp.	Paramacrochiron sp.	Present from Jan–Mar	?	O, SUM	Kolambugan, Lanao del Norte	HC	N	2013 Dec– 2015 Jul	NS	Boco, Metillo & Papa, 2014	
Cephea cephea Forskål, 1775	Alepas pacifica Pilsbry, 1907	Barnacles 44 mm wide present on umbrella and oral arms. Additional details absent	?	B, O	Japanese Coast	?	?	?	?	Hiro, 1937 via Pagès, 2000	
Lobonema sp.	Callinectes sp.	Instar 1 cm	MG, I	?	Gulf of Tehuantepec	?	?	?	?	Bieri unpubl. data via Towanda & Thuesen, 2006	
Lobonemoides robustus Stiasny, 1920	Charybdis feriata Linnaeus, 1758	Present in Gulf of Thailand from July to October as well	MG, J	?	Carigara Bay, Leyte Island	HC	L	2013 23 August	NS	Kondo et al., 2014	
Lychnorhiza lucerna Haeckel, 1880	Cyrtograpsus affinis Dana, 1851	N/A	A	SG	Rio de la Plata Estuary	TR	N	2006 Mar	?	Schiariti et al., 2012	
Lychnorhiza lucerna Haeckel, 1880	Grapsoidea gn sp.	N/A	J	?	Cananéia, Brazil	TR	L	2013 Feb-2014 May	5–15m	Gonçalves et al., 2016	
Lychnorhiza lucerna Haeckel, 1880	Leander paulensis Ortmann, 1897	N/A	M	?	Cananéia, Brazil	TR	L	2013-2014	5–15m	Gonçalves et al., 2016	
Lychnorhiza lucerna Haeckel, 1880	Libinia dubia de Brito Capello, 1871	40% of individuals were living on medusae, all juveniles were living on medusae	M, F, OF, J	O, SUB, B	Cananéia, Brazil	TR	N	2012 Jul	5–15 m	Gonçalves et al., 2017	
Lychnorhiza lucerna Haeckel, 1880	Libinia ferreirae de Brito Capello, 1871	N/A	F, M, J	?	Cananéia and Rio de Janeiro state, Macaé	TR	N	2013–2014	5–15m	Gonçalves et al., 2016	
Lychnorhiza lucerna Haeckel, 1880	Libinia ferreirae de Brito Capello, 1871	N/A	?	SUM, O	Maranhão state	HC	N	2005–2006 Mar	?	de Andrade Santos, Feres & Lopes, 2008	
Lychnorhiza lucerna Haeckel, 1880	Libinia ferreirae de Brito Capello, 1871	Young crabs, transport and protection	J, F, M	SG, O	State of Paraná	TR	N	1997–2004 All yr	8–30 m	Nogueira Júnior & Haddad, 2005	
Lychnorhiza lucerna Haeckel, 1880	Libinia spinosa Guérin, 1832	N/A	F	?	Ubatuba	TR	N	2013 Jul–2014 Aug	5–15m	Gonçalves et al., 2016	
Lychnorhiza lucerna Haeckel, 1880	Libinia spinosa Guérin, 1832	Dispersion, protection and food particulate theft	?	?	Rio del Plata	MULTI	N	2007 Jan-Mar	?	Schiariti et al., 2012	
Lychnorhiza lucerna Haeckel, 1880	Libinia spinosa Guérin, 1832	Dispersion and food particulate theft, Jan-Feb	?	?	Punta del Este	?	?	Jan-Feb	?	Vaz-Ferreira, 1972 via Schiariti et al., 2012	
Lychnorhiza lucerna Haeckel, 1880	Libinia spinosa Guérin, 1832	Transportation and food theft, no more than two crabs/medusa	?	SG	Mar Chiquita Estuary	?	L	?	NS	Zamponi, 2002 via Schiariti et al., 2012	
Lychnorhiza lucerna Haeckel, 1880	Periclimenes paivai Chace, 1969	72% of collected medusae had associate	MG, F, OF, J	SUM	Paraíba River estuary	HC	N	2016 Apr	NS	Baeza et al., 2017	
Lychnorhiza lucerna Haeckel, 1880	Periclimenes paivai Chace, 1969	N/A	OF	SUM	Sao Paolo	TR		2012 Sep–Oct	5–15m	de Moraes et al., 2017	
Lychnorhiza lucerna Haeckel, 1880	Periclimenes paivai Chace, 1969	N/A	OF, M	?	Cananéia	TR	N	2013–2014	5–15m	Gonçalves et al., 2016	
Lychnorhiza lucerna Haeckel, 1880	Periclimenes sp.	Facultative commensal, feeding on mucus, large proportion ovigerous females	OF, A, J	SUM	São Paulo state	HC	N	1999–2002, 2005 Aug + 2006 Jul	NS	Filho et al., 2008	
Lychnorhiza lucerna Haeckel, 1880	Synidotea marplatensis Giambiagi, 1922	N/A	?	SG, O, B	Guaratuba, Paraná e Barra do Saí, Santa Catarina	TR	L	2003–2004 Aug–Dec	8–14 m	Nogueira Junior & Silva (2005)	
Lychnorhiza malayensis Stiasny, 1920	Paramacrochiron sewelli Reddiah, 1968	100 + epibionts from 5 hosts	F, M	?	Ennore estuary near Madras	HC	L	1964 Apr	?	Reddiah, 1968	
Mastigias papua Lesson, 1830	Chlorotocella gracilis Balss, 1914	Collected from ten medusae	M, F, OF	O	Tanabe Bay, Japan	?	N	1965 Oct	?	Hayashi & Miyake, 1968	
Mastigias papua Lesson, 1830	Latreutes anoplonyx Kemp, 1914	Collected from ten medusae	M, F, OF	O	Tanabe Bay, Japan	?	N	1965 Oct	?	Hayashi & Miyake, 1968	
Mastigias papua Lesson, 1830	Latreutes mucronatus Stimpson, 1860	Collected from ten medusae	M, F, OF	O	Tanabe Bay, Japan	?	N	1965 Oct	?	Hayashi & Miyake, 1968	
Nemopilema nomurai Kishinouye, 1922	Alepas pacifica Pilsbry, 1907	Substrate	M, F, OF	B	Western Coast of Japan	HC	N	2005–2009	?	Yusa et al., 2015	
Nemopilema nomurai Kishinouye, 1922	Charybdis feriata Linnaeus, 1758	5 juveniles present on one host on the oral arms, one adult present under the bell of a second medusa.	J & M	O, SUM	Mirs Bay, Hong Kong	?	L	1970 Oct	?	Trott, 1972	
Nemopilema nomurai Kishinouye, 1922Netrostoma setouchianum Kishinouye, 1902	Latreutes anoplonyx Kemp, 1914	Exhibits hiding behavior	M, F, OF	O, SUB	Miyazu and Sanriku, Japan	OBS. HC, SC	L	2003 Nov	?	Hayashi, Sakagami & Toyoda, 2004	
Netrostoma setouchianum Kishinouye, 1902	Chlorotocella gracilis Balss, 1914	Single specimen	?	O	Seto Inland Sea, Japan	HC	L	2010 Sep	NS	Ohtsuka et al., 2011	
Netrostoma setouchianum Kishinouye, 1902	Latreutes mucronatus Stimpson, 1860	Mix of sexes and ages of epibiont from two host individuals, 7 on one and 54 epibionts on the other	M, F, OF, J	O	Seto Inland Sea, Japan	HC	L	2010 Sep	NS	Ohtsuka et al., 2011	
Phyllorhiza punctata von Lendenfeld, 1884	Charybdis feriata Linnaeus, 1758	Single specimen from August 2014	MG	?	Various bays, Philippines	HC	L	2014–2015, Feb–Apr	NS	Boco & Metillo, 2018	
Phyllorhiza punctata von Lendenfeld, 1884	Latreutes anoplonyx Kemp, 1914	N/A	OF, A	B	NT Australia	HC	L	1993	NS	Bruce, 1995	
Phyllorhiza punctata von Lendenfeld, 1884	Libinia ferreirae de Brito Capello, 1871	Feb–Jul	??	SUM	Sao Paulo	?	?	Feb-Jul	?	Moreira, 1961 via Schiariti et al., 2012	
Phyllorhiza punctata von Lendenfeld, 1884	Paramacrochiron sp.	Two specimens from Leyte Gulf- Guiuan in April 2015	?	?	Various bays, Philippines	HC	L	2014–2015, Feb–Apr	NS	Boco & Metillo, 2018	
Pseudorhiza haeckeli Haacke, 1884	Cymodoce gaimardii H. Milne Edwards, 1840	N/A	?	?	Port Phillip Bay,Victoria	HC	N	2011 Sep + 2012 Feb	NS	Browne, 2015	
Pseudorhiza haeckeli Haacke, 1884	Hyperia gaudichaudii H. Milne Edwards, 1840	Exhibit cradle positioning for filter feeding	?	EX	Port Phillip Bay,Victoria	HC	N	2009 Sep + 2012 Feb	NS	Browne, 2015	
Pseudorhiza haeckeli Haacke, 1884	Themisto australis Stebbing, 1888	N/A	?	?	Port Phillip Bay,Victoria	HC	N	2010 Sep + 2012 Feb	NS	Browne, 2015	
Rhizostoma pulmo Macri, 1778	Hyperia galba Montagu, 1813	Peak in Oct, preference for mature medusae, consume host gonad	J, A	O	German Bight	HC + SC	?	1984–1985	?	Dittrich, 1988	
Rhizostoma pulmo Macri, 1778	Iphimedia eblanae Spence Bate, 1857	Present in the brachial cavities, mouthpart shape leads to speculation that these are semi-parasitic short-term associates	?	GVC	Dublin Bay, Ireland	?	N		NS	Bate, 1862 via Vader, 1972	
Rhizostoma sp.	Latreutes anoplonyx Kemp, 1914	N/A	?	?	Indonesia	?	?	?	?	Hayashi, Sakagami & Toyoda, 2004	
Rhizostoma sp.	Paramacrochiron rhizostomae Reddiah, 1968	N/A	F, M, J	?	Vaalai Island, Madras State	HC	L	1967 Mar	NS	Reddiah, 1968	
Rhizostomatidae gn. sp.	Alepas pacifica Pilsbry, 1907	2 barnacles on the umbrellar margin up to 68 mm in length	?	MA	Morrison Bay, Mergui Arch	?	L	1914	NS	Annandale, 1914 via Pagès, 2000	
Rhopilema esculentum Kishinouye, 1891	Charybdis feriata Linnaeus, 1758	Juvenile transport	J	O	Sagami Bay	?	?	October	?	Suzuki, 1965 via Pagès, 2000	
Rhopilema esculentum Kishinouye, 1891	Latreutes anoplonyx Kemp, 1914	N/A	?	?	Northeast China	?	?	?	?	Hayashi, Sakagami & Toyoda, 2004	
Rhopilema hispidum Vanhöffen, 1888	Charybdis annulata Fabricius, 1798	N/A	??	SUM	Palk Bay, Sri Lanka	?	L	1950 Jul	?	Panikkar & Raghu Prasad, 1952 via Towanda & Thuesen, 2006	
Rhopilema hispidum Vanhöffen, 1888	Charybdis feriata Linnaeus, 1758	Present on all medusae collected in Aug	J & MG	?	Panguil Bay	HC	N	2014 Feb+Aug	NS	Boco & Metillo, 2018	
Rhopilema hispidum Vanhöffen, 1888	Hippolytidae gn sp.	Three associates on a single medusa from Feb	?	?	Panguil Bay	HC	L	2014 Feb+Aug	NS	Boco & Metillo, 2018	
Rhopilema hispidum Vanhöffen, 1888	Latreutes sp. aff. anoplonyx Kemp, 1914	N/A	??	MA, O	Kukup, Malaysia	?	L	2009 Mar + Oct	?	Ohtsuka et al., 2010	
Rhopilema hispidum Vanhöffen, 1888	Latreutes sp. aff. anoplonyx Kemp, 1914	N/A	??	?	Sichang Island, Thailand	?	L	2009 Oct	?	Ohtsuka et al., 2010	
Rhopilema hispidum Vanhöffen, 1888	Paramacrochiron sp.	On 67% of medusae from Aug collection	?	?	Panguil Bay	HC	L	2014 Feb+Aug	NS	Boco & Metillo, 2018	
Rhopilema hispidum Vanhöffen, 1888	Paramacrochiron sp.	Theorized ectoparasite, no record of actual consumption.	A & L	O	Laem Phak Bia, Thailand	HC	L	2010 Oct	NS	Ohtsuka, Boxshall & Srinui, 2012	
Rhopilema nomadica Galil, Spanier & Ferguson, 1990	Charybdis feriata Linnaeus, 1758	Many hosts containing multipe associations, only some possess Charybdis, never more than one crab per medusa.	?	O, SUB	Delagoa Bight, Mozambique	HC	L	1988 Mar + 1992 Mar	NS	Berggren, 1994	
Rhopilema nomadica Galil, Spanier & Ferguson, 1990	Periclimenes nomadophila Berggren, 1994	Many hosts containing multipe associations	F, OF, M	O, SUB	Delagoa Bight, Mozambique	HC	N	1988 Mar + 1992 Mar	NS	Berggren, 1994	
Rhopilema sp.	Conchoderma virgatum Spengler, 1789	22 barnacles on the umbrellar Margin (ex and sub) on host of 320 mm diameter	?	MA	Tranquebar, Bengala Gulf	?	L	?	?	Fernando & Ramamoorthi, 1974 via Pagès, 2000	
Stomolophus meleagris, Agassiz, 1860	Charybdis feriata Linnaeus, 1758	N/A	F & J	O	Hong Kong	?	?	?	?	Morton, 1989 via Towanda & Thuesen, 2006	
Stomolophus meleagris, Agassiz, 1860	Conchoderma cf virgatum Spengler, 1789	Mature jellyfish, scarring and lesions around attachment site	?	B	Gulf of California	HC	L	2010 Apr	NS	Álvarez-Tello, López-Martínez & Rodríguez-Romero, 2013	
Stomolophus meleagris, Agassiz, 1860	Libinia dubia H. Milne Edwards, 1834	All medusa harbored crabs, no more than one crab per medusa	A	SUM	Murrell’s Inlet, SC	?	N	1927 May	“relatively deep”	Corrington, 1927	
Stomolophus meleagris, Agassiz, 1860	Libinia dubia H. Milne Edwards, 1834	N/A	?	SUM	Beaufort, NC	TR	N	1927 Jul–Oct	NS	Gutsell, 1928	
Stomolophus meleagris, Agassiz, 1860	Libinia dubia H. Milne Edwards, 1834	Juvenile associations, parasitic, transient	J	W	Mississippi sound	HC	N	1968 Jul–Oct	NS	Phillips, Burke & Keener, 1969	
Stomolophus meleagris, Agassiz, 1860	Libinia dubia H. Milne Edwards, 1834	Highly variable seasonally, high in July, low in Dec	F, M, J	O, MA	Wrightsville Beach Jetty NC	HC	N	1983 May–Dec	NS	Rountree, 1983	
Stomolophus meleagris, Agassiz, 1860	Libinia dubia H. Milne Edwards, 1834	Feeding	?	EXC	Onslow Bay, NC	SC	?	??	?	Shanks & Graham, 1988 via Schiariti et al., 2012	
Stomolophus meleagris, Agassiz, 1860	Libinia dubia H. Milne Edwards, 1834	N/A	?	?	Indian River Lagoon, Florida	HC	?	2003 Mar	?	Tunberg & Reed, 2004	
Stomolophus meleagris, Agassiz, 1860	Penaeus stylirostris Stimpson, 1871	N/A	?	?	Malaga Bay, Colombia	HC	?	2015 Nov + 2017 Apr	NS	Riascos et al., 2018	
Thysanostoma thysanura Haeckel, 1880	Paramacrochiron sp.	N/A	?	?	Sirahama	?	?	1969	?	Humes, 1970	
Versuriga anadyomene Maas, 1903	Charybdis feriata Linnaeus, 1758	Large medusae	?	?	Leyte Gulf- Guiuan	HC	L	2014–2015, Feb–Apr	NS	Boco & Metillo, 2018	
Versuriga anadyomene Maas, 1903	Charybdis feriata Linnaeus, 1758	N/A	??	SUM	Pari Island, Indonesia	?	L	2009 Nov	?	Ohtsuka, Boxshall & Srinui, 2012	
Versuriga anadyomene Maas, 1903	Latreutes anoplonyx Kemp, 1914	N/A	A & J	SUM	NT Australia	HC	L	1993	NS	Bruce, 1995	
Versuriga anadyomene Maas, 1903	Paramacrochiron sp.	Large medusae	?	?	Leyte Gulf- Guiuan	HC	N	2014–2015, Feb–Apr	NS	Boco & Metillo, 2018	
Semaeostomeae											
Aurelia aurita Linnaeus, 1758	Hyperia galba Montagu, 1813	N/A	A & J & OF	?	Narragansett Marine Laboratory	HC	?	1955 June	NS	Bowman, Meyers & Hicks, 1963	
Aurelia aurita Linnaeus, 1758	Hyperia galba Montagu, 1813	Preference for mature medusae, infestation increases as gonads develop, peak in Oct, consume host gonad	J, A	O	German Bight	HC + SC		1984–1985	?	Dittrich, 1988	
Aurelia aurita Linnaeus, 1758	Libinia dubia H. Milne Edwards, 1834	Eating medusa tissue, residence within bell, excavation behaviors 19.9% of medusae examined 300-500 m from shore had phyllosoma, none on Aurelia near shore, likely parasitoid.	?	EXC	Chesapeake Bay	?	?	1963 Aug	?	Jachowski, 1963	
Aurelia aurita Linnaeus, 1758	Scyllarus sp.	Riding small medusae, pierced exumbrella with pereiopods	PL	EX	Bimini, Bahamas	HC	N	1973 Oct	NS	Herrnkind, Halusky & Kanciruk, 1976	
Aurelia coerulea von Lendenfeld, 1884	Ibacus ciliatus von Siebold, 1824	February to May, 97.6% female, largely one female per host, occasionally M/F pair, 1/3 of parasites were ovigerous.	PL	EX	Yamaguchi, Japan	OBS	L	?	?	Wakabayashi, Tanaka & Abe, 2017 via Wakabayashi, Tanaka & Phillips, 2019	
Aurelia coerulea von Lendenfeld, 1884	Oxycephalus clausi Bovallius, 1887	No breakdown by specific host	OF, F	EX	Nagato, Yamaguchi, Japan	OBS	N	2012-2018	0–5 m	Mazda et al., 2019	
Aurelia limbata Brandt, 1835	Hyperia galba Montagu, 1813	N/A	F, J	O	Okirai Bay	?	L	2009 Apr	?	Ohtsuka et al., 2010	
Aurelia sp.	Nitokra medusaea Humes, 1953	Engage in excavation, many epibionts on a single 5′ medusa	F, M, OF	EXC	New Hampshire coast	HC	L	1952	NS	Humes, 1953	
Chrysaora colorata Russell, 1964	Latreutes anoplonyx Kemp, 1914	N/A	?	?	Kuwait Bay	TR	?	1981 Sept–1982 Aug	?	Grabe & Lees, 1995	
Chrysaora colorata Russell, 1964	Metacarcinus gracilis Dana, 1852	Dispersion, protection and feeding, Mar–Aug	MG	?	Monterey Bay	?	?	1991/1992 Mar–Aug	?	Graham, 1989 via Schiariti et al., 2012	
Chrysaora colorata Russell, 1964	Metacarcinus gracilis Dana, 1852	Early stages of crabs on medusae	J, MG	?	Califorina	?	?	?	?	Wrobel & Mills, 1998 via Schiariti et al., 2012	
Chrysaora fuscescens Brandt, 1835	Cancer sp.	Crabs gain dispersion	?	?	Monterey Bay	?	?	?	?	Graham, 1994 via Schiariti et al., 2012	
Chrysaora fuscescens Brandt, 1835	Hyperoche medusarum Kröyer, 1838	Infestations occur in late summer	?	?	NE Pacific, Oregon and northern California	?	?	?	?	Larson, 1990	
Chrysaora fuscescens Brandt, 1835	Metacarcinus gracilis Dana, 1852	N/A	?	?	NE Pacific “off California”	?	?	?	?	Larson, 1990	
Chrysaora hysoscella Linnaeus, 1767	Hyperia galba Montagu, 1813	Peak in Oct, reference for mature medusae, consume host gonad	J, A	O	German Bight	HC + SC		1984–1985	?	Dittrich, 1988	
Chrysaora lactea Eschscholtz, 1829	Brachyscelus cf. rapacoides Stephensen, 1925	Parasite	L, J	W, O	Sao Sebastian Channel	TR	L	2015 Nov	?	Puente-Tapia et al., 2018	
Chrysaora lactea Eschscholtz, 1829	Cymothoa catarinensis Thatcher, Loyola e Silva, Jost & Souza-Conceiçao, 2003	N/A	?	EX	Guaratuba, Paraná e Baía Norte, Florianópolis, Santa Catarina	TR	L	2003 + 2005, Nov + May	8–14 m	Nogueira Junior & Silva, 2005	
Chrysaora lactea Eschscholtz, 1829	Periclimenes sp.	Facultative commensal, feeding on mucus, large proportion ovigerous females	OF, A, J	SUM	São Paulo state	HC	?	1999–2002 + 2006 Jul	NS	Filho et al., 2008	
Chrysaora lactea Eschscholtz, 1829	Synidotea marplatensis Giambiagi, 1922	N/A	?	SUM	Guaratuba, Paraná e Barra do Saí, Santa Catarina,	TR	L	2003–2004 Aug–Dec	8–14 m	Nogueira Junior & Silva, 2005	
Chrysaora melanaster Brandt, 1835	Hyperia galba Montagu, 1813	N/A	J	SUM, O	Takehara City (34 18′N, 132 55′E)	?	L	2009 Apr + Jun	?	Ohtsuka, Boxshall & Srinui, 2012	
Chrysaora pacifica Goette, 1886	Oxycephalus clausi Bovallius, 1887	February to May, 97.6% female, largely one female per host, occasionally M/F pair, 1/3 of parasites were ovigerous. No breakdown by specific host	OF, F	EX	Nagato, Yamaguchi, Japan	OBS	L	2012–2018	0–5 m	Mazda et al., 2019	
Chrysaora plocamia Lesson, 1830	Hyperia curticephala Vinogradov & Semenova, 1985	Mean 0f 174. 4 amphipods/host, 79% female, ingested mesoglea	M, F, OF	W	Mejillones Bay	SC	N	2005 Feb	NS	Oliva, Maffet & Laudien, 2010	
Chrysaora quinquecirrha Desor, 1848	Callinectes sapidus Rathbun, 1896	Not feeding on medusa	??	EX	Mississippi sound	HC	L	1968 Aug	NS	Phillips, Burke & Keener, 1969	
Chrysaora quinquecirrha Desor, 1848	Libinia dubia H. Milne Edwards, 1834	Lower incidence rate near surface than bottom trawls, actively feeding on medusae	??	B, O	Mississippi sound	MULTI	N	1968 Aug	NS	Phillips, Burke & Keener, 1969	
Chrysaora quinquecirrha Desor, 1848	Pseudomacrochiron stocki Sars, 1909	12 specimens from 10 hosts	F, M	?	Madras Marina	HC	N	1967, Oct	?	Reddiah, 1969	
Chrysaora sp.	Cancer sp. cf. antennarius*	N/A	J, MG	?	Southern California Bight	HC	N	1989 Jul–Sep	NS	Martin & Kuck, 1991	
Chrysaora sp.	Hyperia medusarum Müller, 1776	N/A	F	?	Southern California Bight	HC	L	1989. Jul–Sep	NS	Martin & Kuck, 1991	
Chrysaora sp.	Metamysidopsis elongata Holmes, 1900	N/A	M	?	Southern California Bight	HC	L	1989. Jul–Sep	NS	Martin & Kuck, 1991	
Chrysaora sp.	Mysidopsis cathengelae Gleye, 1982	N/A	M	?	Southern California Bight	HC	L	1989. Jul–Sep	NS	Martin & Kuck, 1991	
Cyanea capillata Linnaeus, 1758	Alepas pacifica Pilsbry, 1907	Seven barnacles from 14.5-37 mm in length on the exumbrella and umbrellar Margin.	?	MA, EX	Marion Bay, Tazmania	?	L	1985	?	Liu & Ren, 1985 via Pagès, 2000	
Cyanea capillata Linnaeus, 1758	Hyperia galba Montagu, 1813	Inverted positioning, plentiful in the spring	A & J & OF	MA, EX	Narragansett Marine Laboratory	HC	N	1954 Sep –1955 Aug	NS	Bowman, Meyers & Hicks, 1963	
Cyanea capillata Linnaeus, 1758	Hyperia galba Montagu, 1813	N/A	A & J & OF	?	Niantic River	TR	N	1960, May + Jun	NS	Bowman, Meyers & Hicks, 1963	
Cyanea capillata Linnaeus, 1758	Hyperia galba Montagu, 1813	Peak in Oct, reference for mature medusae, consume host gonad	J, A	O	German Bight	HC + SC		1984–1985	?	Dittrich, 1988	
Cyanea capillata Linnaeus, 1758	Hyperoche medusarum Kröyer, 1838	Single specimen in May	J	?	Niantic River	HC	L	1960, May + Jun	NS	Bowman, Meyers & Hicks, 1963	
Cyanea capillata Linnaeus, 1758	Themisto australis Stebbing, 1888	Cradle positioning, no bell damage, all sampled epibionts submature females	JF	EX	Rye Pier (38°23′S, 144°50′E)	HC	N	1995, Jun–Oct	NS	Condon & Norman, 1999	
Cyanea nozakii Kishinouye, 1891	Alepas pacifica Pilsbry, 1907	Relationship uncharacterized except to note epibiont presence on umbrella and oral arms	?	B, O	Japanese Coast	?	?	?	?	Hiro, 1937 via Pagès, 2000	
Cyanea nozakii Kishinouye, 1891	Alepas pacifica Pilsbry, 1907	3 barnacles on the umbrella up to a length of 130 mm	?	EX	Shanghai	?	?	1946	?	Tubb, 1946 via Pagès, 2000	
Cyanea nozakii Kishinouye, 1891	Alepas pacifica Pilsbry, 1907	Substrate	M, F, OF	B	Western Coast of Japan	HC	L	2005–2009	?	Yusa et al., 2015	
Deepstaria enigmatica Russell, 1967	Anuropidae gn. sp.	Two anuropids close to the oral arm base on one medusa	?	O, SUM	Mutsu Bay	ROV	L	2002 Apr/May	669 m	Lindsay et al., 2004	
Deepstaria enigmatica Russell, 1967	Anuropus sp.	Parasitic	?	SUM	San Diego Trough	ROV	L	1966 Oct	723 m	Barham & Pickwell, 1969	
Diplulmaris malayensis Stiasny, 1935	Alepas pacifica Pilsbry, 1907	15 barnacles found on 10 hosts, mostly attached to the subumbrellar margins. 1 to 3 epibionts per host. 11 were oriented towards the GVC opening and oral arms of the host. Hypothesized consumption of gonadal tissue by this epibiont	?	MA	34 29.4′N, 138 32.6′E	TR	N	1981 Jun	NS	Pagès, 2000	
Pelagia noctiluca Forsskål, 1775	Alepas pacifica Pilsbry, 1907	Over 100 barnacles on the umbrellar and oral arm regions of an unknown number of medusae	?	B, O	Japanese Coast	?	?	?	?	Hiro, 1937 via Pagès, 2000	
Pelagia noctiluca Forsskål, 1775	Alepas pacifica Pilsbry, 1907	N/A	?	SUM	39N, 52W	?	?	?	?	Madin unpubl data via Pagès, 2000	
Pelagia noctiluca Forsskål, 1775	Alepas pacifica Pilsbry, 1907	One barnacle 20 mm long, present on an oral arm	?	O	Misaki, Japan	?	L	?	?	Utinomi, 1958 via Pagès, 2000	
Pelagia noctiluca Forsskål, 1775	Anelasma sp.	Medusae up to 60 mm in diameter, unknown epibiont number, size and position.	?	?	Kuroshio, Japan	?	?	?	?	Kishinouye, 1902 via Pagès, 2000	
Pelagia noctiluca Forsskål, 1775	Oxycephalus clausi Bovallius, 1887	February to May, 97.6% female, largely one female per host, occasionally M/F pair, 1/3 of parasites were ovigerous. No breakdown by specific host	OF, F	EX	Nagato, Yamaguchi, Japan	OBS	L	2012–2018	0–5 m	Mazda et al., 2019	
Pelagia noctiluca Forsskål, 1775	Thamneus rostratus Bovallius, 1887	Relatively rare species	A & J	SUM	Gulf of California	SC	L	2003 Mar	10 m	Gasca & Haddock, 2004	
Pelagia panopyra Péron & Lesueur, 1810	Ibacus sp.	Each medusa had a phyllosoma larva firmly attached to the bell surface. The larvae were difficult to remove without injuring them, considered parasitoid relationship	PL	EX	Sydney Harbor	?	L	1960 May	?	Thomas, 1963	
Phacellophora camtschatica Brandt, 1835	Alepas pacifica Pilsbry, 1907	2 5–5.1 cm long barnacles on a 50 mm	?	?	Tasman sea	?	L	1968	?	Utinomi, 1968 via Pagès, 2000	
Phacellophora camtschatica Brandt, 1835	Hyperia medusarum Müller, 1776	Parasitoid, May to Sept, 100s of amphipods, 100% of hosts had infestation in July	M & F & J	O	Puget Sound	HC	N	1994-2003 May-Oct	NS	Towanda & Thuesen, 2006	
Phacellophora camtschatica Brandt, 1835	Metacarcinus gracilis Dana, 1852	Association appears in May, once bell widths of hosts begin to exceed 3 cm, peaks in June/July, few after mid-Oct	MG & I	B, O	Puget Sound	HC	N	1994–2003 May–Oct	NS	Towanda & Thuesen, 2006	
Poralia rufescens Vanhöffen, 1902	Lanceola clausii Bovallius, 1885	N/A	F, M, J	SUM	Suruga Bay	ROV	L	2002 Apr	867–1,697 m	Hughes & Lindsay, 2017	
Poralia rufescens Vanhöffen, 1902	Lysianassinae gn sp.	Attached at base of oral arms, 1–6 per medusa	?	O, SUM	Japan Trench	ROV	N	2002 Apr/May	500–1000 m	Lindsay et al., 2004	
Poralia rufescens Vanhöffen, 1902	Pseudocallisoma coecum Holmes, 1908	Only juvenile specimens	J	O	Japan Trench	ROV	L	2002 Apr–May	576–732 m	Hughes & Lindsay, 2017	
Hydrozoa											
Anthoathecata											
Bythotiara depressa Naumov, 1960	Scina sp.	N/A	?	?	Gulf of California	ROV	L	2007 Dec	494 m	Gasca, Hoover & Haddock, 2015	
Bythotiara sp.	Mimonectes sphaericus Bovallius, 1885	N/A	?	B	Gulf of California	ROV	L	2006 May	690 m	Gasca, Hoover & Haddock, 2015	
Leuckartiara octona Fleming, 1823	Hyperia medusarum Müller, 1776	N/A	JM	?	Gulf of California	SC	L	2006 Sep	<30 m	Gasca, Hoover & Haddock, 2015	
Leuckartiara zacae Bigelow, 1940	Hyperia medusarum Müller, 1776	N/A	F, J	?	Monterey California	SC	L	2004 May	10 m	Gasca, Suárez-Morales & Haddock, 2007	
Leuckartiara zacae Bigelow, 1940	Lestrigonus schizogeneios Stebbing, 1888	N/A	JF	?	Monterey California	SC	L	2004 May	5–15m	Gasca, Suárez-Morales & Haddock, 2007	
Neoturris sp.	Hyperia medusarum Müller, 1776	N/A	OF, J	?	Monterey California	ROV	L	2004 May	237 m	Gasca, Suárez-Morales & Haddock, 2007	
Leptothecata											
Aequorea coerulescens Brandt, 1835	Brachyscelidae gn sp.	N/A	J	?	Gulf of California	SC	L	2003 Mar	10 m	Gasca & Haddock, 2004	
Aequorea coerulescens Brandt, 1835	Brachyscelus crusculum Spence Bate, 1861	N/A	JM, A & OF	EX	Gulf of California	SC	L	2003 Mar	10–15 m	Gasca & Haddock, 2004	
Aequorea coerulescens Brandt, 1835	Ibacus ciliatus von Siebold, 1824	N/A	PL	?	Yamaguchi, Japan	?	?	?	?	Wakabayashi, Tanaka & Abe, 2017 via Wakabayashi, Tanaka & Phillips, 2019	
Aequorea coerulescens Brandt, 1835	Oxycephalus clausi Bovallius, 1887	February to May, 97.6% female, largely one female per host, occasionally M/F pair, 1/3 of parasites were ovigerous. No account breakdown by specific host	OF, F	EX	Nagato, Yamaguchi, Japan	OBS	N	2012–2018	0–5 m	Mazda et al., 2019	
Aequorea coerulescens Brandt, 1835	Sapphirina nigromaculata Claus, 1863	N/A	?	MA	Gulf of California	SC	L	2003 Mar	10 m	Gasca & Haddock, 2004	
Aequorea coerulescens Brandt, 1835	Thamneus rostratus Bovallius, 1887	Relatively rare amphipod species	J	B	Gulf of California	SC	L	2003 Mar	10 m	Gasca & Haddock, 2004	
Aequorea eurodina* Péron & Lesueur, 1810	Hyperia gaudichaudii H. Milne Edwards, 1840	2 attached to one medusa	?	?	Port Phillip Bay, Australia	HC	L	2009 Sep + 2012 Feb	NS	Browne, 2015	
Aequorea macrodactyla Brandt, 1835	Ibacus novemdentatus Gibbes, 1850	N/A	PL	?	Nagasaki, Japan	?	?	?	?	Shojima, 1973 via Wakabayashi, Tanaka & Phillips, 2019	
Aequorea victoria Murbach & Shearer, 1902	Ibacus ciliatus von Siebold, 1824	Riding small medusae, pierced exumbrella with pereiopods, attached to a salp as well, parasitoid relationship hypothesized	PL	EX	Japan	OBS	L	?	?	Wakabayashi, Tanaka & Phillips, 2019	
Chromatonema erythrogonon, Bigelow, 1909	Hyperoche medusarum Kröyer, 1838	N/A	OF	?	Gulf of California	ROV	L	2003 Mar	1,100 m	Gasca & Haddock, 2004	
Clytia hemisphaerica Linnaeus, 1767	Eduarctus martensii Pfeffer, 1881	N/A	PL	?	Yamaguchi, Japan	?	?	?	?	Wakabayashi, Tanaka & Abe, 2017 via Wakabayashi, Tanaka & Phillips, 2019	
Clytia sp.	Metopa borealis G. O. Sars, 1883	Association from Oct to March, epibionts passed between medusae	?	B, O	West Scotland	?	N	Oct–Mar	?	Elmhirst, 1925 via Vader, 1972	
Eutonina indicans Romanes, 1876	Tryphana malmii Boeck, 1871	N/A	?	?	Gulf of California	ROV	L	2006 May	202 m	Gasca, Hoover & Haddock, 2015	
Mitrocoma cellularia Agassiz, 1862	Hyperoche medusarum Kröyer, 1838	N/A	OF, J	W	Monterey California	SC	L	2004 May	10 m	Gasca, Suárez-Morales & Haddock, 2007	
Mitrocoma cellularia Agassiz, 1862	Tryphana malmii Boeck, 1871	N/A	JF		Monterey California	SC	L	2004 May	5-15m	Gasca, Suárez-Morales & Haddock, 2007	
Tima bairdii Johnston, 1833	Metopa alderi Spence Bate, 1857	Speculates year-round relationship, mobile on medusa, did not feed on host tissue, fed on mucus	J & A & OF	SUM, O, B, T	Bergen	?	N	1970 Apr	?	Vader, 1972	
Tima formosa Agassiz, 1862	Hyperoche medusarum Kröyer, 1838	N/A	JF	?	Narragansett Marine Laboratory	HC	L	1954 Sep– 1957 Aug	NS	Bowman, Meyers & Hicks, 1963	
Tima sp.	Iulopis mirabilis Bovallius, 1887	N/A	J & A	?	Gulf of California	SC	L	2006 Sep	<30 m	Gasca, Hoover & Haddock, 2015	
Limnomedusae											
Liriope tetraphylla Chamisso & Eysenhardt, 1821	Simorhynchotus antennarius Claus, 1871	N/A	0F	?	Gulf of California	SC	L	2006 Jun	<30 m	Gasca, Hoover & Haddock, 2015	
Liriope tetraphylla Chamisso & Eysenhardt, 1821	Ibacus ciliatus von Siebold, 1824	N/A	PL	?	Nagasaki, Japan	?	?	?	?	Shojima, 1973 via Wakabayashi, Tanaka & Phillips, 2019	
Liriope sp.	Scyllarus chacei Holthuis, 1960	30% of phyllosoma attached to at least one GZ species, primarily hydrozoa, parasitoid relationship	PL	EX	Northern Gulf of Mexico	OBS,
TR	N	2015 Oct	1–31 m	Greer et al., 2017	
Olindias sambaquiensis Müller, 1861	Brachyscelus cf. rapacoides Stephensen, 1925	Reduction in mouthpart of epibionts higher in females	J	?	Sao Sebastian Channel	TR	L	2015 Nov	?	Puente-Tapia et al., 2018	
Olindias sambaquiensis Müller, 1861	Synidotea marplatensis Giambiagi, 1922	N/A	?	EX	Guaratuba, Paraná e Barra do Saí, Santa Catarina,	TR	L	2003–2004 Aug-Dec	8–14 m	Nogueira Junior & Silva, 2005	
Narcomedusae											
Aegina citrea Eschscholtz, 1829	Iulopis loveni Bovallius, 1887	N/A	F	?	Gulf of California	ROV	L	2007 Jan	83 m	Gasca, Hoover & Haddock, 2015	
Aegina citrea Eschscholtz, 1829	Iulopis mirabilis Bovallius, 1887	N/A	A	?	Gulf of California	ROV	L	2006 Oct	1,286–1,478 m	Gasca, Hoover & Haddock, 2015	
Aegina citrea Eschscholtz, 1829	Lanceola pacifica Stebbing, 1888	N/A	M		Monterey California	ROV	L	2005 Apr	1,322 m	Gasca, Suárez-Morales & Haddock, 2007	
Aegina citrea Eschscholtz, 1829	Prohyperia shihi Gasca, 2005	N/A	?	?	Gulf of California	ROV	L	2007 Aug	554 m	Gasca, Hoover & Haddock, 2015	
Aegina citrea Eschscholtz, 1829	Pseudolubbockia dilatata Sars, 1909	Refuge and mating, mating pairs with long residence time evident on more than one occasion	M, F	SUM	Monterey California	ROV	L	2004 May	606–1,098 m	Gasca, Suárez-Morales & Haddock, 2007	
Pegantha laevis Bigelow, 1909	Prohyperia shihi Gasca, 2005	N/A	JF	GVC	Gulf of California	ROV	L	2015 Mar	926 m	Gasca & Browne, 2018	
Solmissus incisa Fewkes, 1886	Brachyscelus sp.	N/A	J	?	Gulf of California	ROV	L	2006 May	497 m	Gasca, Hoover & Haddock, 2015	
Solmissus incisa Fewkes, 1886	Thamneus rostratus Bovallius, 1887	N/A	?		Monterey California	ROV	L	2005 Apr	243 m	Gasca, Suárez-Morales & Haddock, 2007	
Solmissus incisa Fewkes, 1886	Tryphana malmii Boeck, 1871	N/A	F		Monterey California	ROV	L	2004 May	458 m	Gasca, Suárez-Morales & Haddock, 2007	
Solmissus incisa Fewkes, 1886	Tryphana malmii Boeck, 1871	N/A	OF	?	Gulf of California	ROV	L	2006 May	295 m	Gasca, Hoover & Haddock, 2015	
Solmissus sp.	Hyperia medusarum Müller, 1776	N/A	JF	?	Gulf of California	ROV	L	2006 Sep	498 m	Gasca, Hoover & Haddock, 2015	
Solmissus sp.	Hyperia sp.	N/A	?	?	Gulf of California	ROV	L	2006 Sep	396–435 m	Gasca, Hoover & Haddock, 2015	
Apolemia sp.	Megalanceoloides aequanime Gasca, 2017	N/A	OF	GVC	Gulf of California	ROV	L	2015 Mar	2,094 m	Gasca & Browne, 2018	
Apolemia sp.	Mimonectes loveni Bovallius, 1885	N/A	F	GVC	Gulf of California	ROV	L	2015 Mar	2,325–2,589 m	Gasca & Browne, 2018	
Athorybia rosacea Forsskål, 1775	Parascelus edwardsi Claus, 1879	Relatively rare amphipod species	?	?	Gulf of California	SC	L	2003 Mar	10 m	Gasca & Haddock, 2004	
Chelophyes appendiculata Eschscholtz, 1829	Paralycaea hoylei Stebbing, 1888	N/A	JF		Monterey California	SC	L	2004 May	5–15m	Gasca, Suárez-Morales & Haddock, 2007	
Diphyes bojani Eschscholtz, 1825	Lestrigonus bengalensis Giles, 1897	N/A	F, JF	W	Cabo Frio (RJ) and the Santa Catarina Island (SC)	TR	L	1980, 17-23 Jan	?	de Lima & Valentin, 2001	
Nectadamas diomedeae Bigelow, 1911	Mimonectes sphaericus Bovallius, 1885	N/A	M		Monterey California	ROV	L	2005 Apr	1,082 m	Gasca, Suárez-Morales & Haddock, 2007	
Nectadamas diomedeae Bigelow, 1911	Mimonectes sphaericus Bovallius, 1885	N/A	J	?	Gulf of California	ROV	L	2006 May	1,344 m	Gasca, Hoover & Haddock, 2015	
Nectadamas diomedeae Bigelow, 1911	Mimonectes stephenseni Pirlot, 1929	N/A	F		Monterey California	ROV	L	2003 May	392 m	Gasca, Suárez-Morales & Haddock, 2007	
Siphonophorae											
Muggiea sp.	Scyllarus chacei Holthuis, 1960	30% of phyllosoma attached to at least one GZ species, primarily hydrozoa, parasitoid relationship hypothesized.	PL	EX	Northern Gulf of Mexico	OBS, TR	N	2015 Oct	1–31 m	Greer et al., 2017	
Physophora hydrostatica Forsskål, 1775	Tryphana malmii Boeck, 1871	N/A	?	?	Gulf of California	ROV	L	2006 Jan	116 m	Gasca, Hoover & Haddock, 2015	
Prayidae gn sp	Scyllaridae gn sp	Attached with pereiopods	PL	EX	Gran Canaria, Spain	OBS	L	1999 Feb	3 m	Ates, Lindsay & Sekiguchi, 2007	
Resomia ornicephala Pugh & Haddock, 2010	Anapronoe reinhardti Stephensen, 1925	N/A	F, JM	?	Gulf of California	ROV	L	2006 Sep	254 m	Gasca, Hoover & Haddock, 2015	
Resomia ornicephala Pugh & Haddock, 2010	Tryphana malmii Boeck, 1871	N/A	OF, A, J	?	Gulf of California	ROV	L	2006 May	204 m	Gasca, Hoover & Haddock, 2015	
Rosacea cymbiformis Delle Chiaje, 1830	Brachyscelus crusculum Spence Bate, 1861	N/A	JF	GVC	Gulf of California	SC	L	2015 Mar	15 m	Gasca & Browne, 2018	
Rosacea cymbiformis Delle Chiaje, 1830	Eupronoe minuta Claus, 1879	N/A	JF	?	Gulf of California	ROV	L	2006 Sep	161 m	Gasca, Hoover & Haddock, 2015	
Rosacea cymbiformis Delle Chiaje, 1830	Paraphronima gracilis Claus, 1879	N/A	J	?	Gulf of California	ROV	L	2006 May	430 m	Gasca, Hoover & Haddock, 2015	
Sulculeolaria quadrivalvis de Blainville, 1830	Simorhynchotus antennarius Claus, 1871	N/A	F	W	Cabo Frio (RJ) and the Santa Catarina Island (SC)	TR	L	1980, 17–23 Jan	?	de Lima & Valentin, 2001	
Trachymedusae											
Haliscera bigelowi Kramp, 1947	Hyperia medusarum Müller, 1776	N/A	J	?	Gulf of California	SC	L	2006 Sep	<30 m	Gasca, Hoover & Haddock, 2015	
Haliscera bigelowi Kramp, 1947	Scina spinosa Vosseler, 1901	N/A	M		Monterey California	ROV	L	2005 Apr	394 m	Gasca, Suárez-Morales & Haddock, 2007	
Haliscera sp.	Scina spinosa Vosseler, 1901	N/A	J	?	Gulf of California	ROV	L	2006 Oct	1,263 m	Gasca, Hoover & Haddock, 2015	
Haliscera sp.	Scina uncipes Stebbing, 1895	N/A	A	?	Gulf of California	ROV	L	2006 May	449 m	Gasca, Hoover & Haddock, 2015	
Pectis tatsunoko Lindsay & Pagès, 2010	Mimonectes spandlii Stephensen & Pirlot, 1931	N/A	JM	SUM	Suruga Bay	ROV	L	2002 Apr	1,967 m	Lindsay & Pagès, 2010	
Notes:

Life Stage and Sex: F, Female; M, Male; MG, Megalopa; A, Adult; E, Egg; J, Juvenile; OF, Ovigerous female; C, Copepodid/Copepodite; I, Instar; PL, Phyllosoma larva

Location on Medusa: EX, Exumbrella; SUM, Subumbrella; O, Oral arms; B, Bell (undifferentiated); GVC, Gastrovascular cavity; SG, Subgenital pit; W, Within medusa (undif.); MA, Umbrellar margin; T, Tentacles

Collection: HC, Hand collection (Nets, buckets, bags, etc.); SC, Scuba and Blue Water Diving; ROV, Remote and Human Operated Vehicles; TR, Boat trawls; MULTI, Multiple methods used; OBS, Observational methods with imaging

Limited Observations: 5 or fewer occurrences catalogued; N, >5 medusae with this epibiont

Depth: NS, Near surface

All: ?, Data missing

Results and discussion

The final table produced by this review process includes 211 recorded interactions between hydrozoan or scyphozoan medusae and crustaceans, extracted from 97 papers (Table 1). For both cnidarians and crustaceans, order, family, genus, and species are included in Supplementary Materials. Results that lacked taxonomic identification (at least Family level) were not included. The final table (Table 1) provides sampling information, such as year and month of sampling, sampling method, and region of sampling. For crustaceans, records include the life stage involved in the interaction, sex of the epibiont, location on the hosts, and additional notes, if available. In most studies, fewer data were available on the cnidarian hosts, reducing the degree to which these interactions could be analyzed in terms of hydromedusan or scyphomedusan life stage. In the next paragraphs, we discuss the jellyfish-crustacea interactions through all of the categories included.

Diversity

Diversity of scyphozoan hosts

A supermajority of records (70%, or 148/211) involves Scyphomedusae, with 53 records involving just the five most common scyphozoan species: Lychnorhiza lucerna (Haeckel, 1880), Catostylus mosaicus (Quoy & Gaimard, 1824), Stomolophus meleagris (Agassiz, 1860), Cyanea capillata (Linnaeus, 1758) and Rhopilema hispidum (Vanhöffen, 1888). These records are heavily concentrated in the upper water column. Deeper water collections (ROV/HOV) were dominated by hydromedusae (69%, or 27/39), while records involving the upper water column (0–30 m) were more common and dominated by scyphomedusae (78%, or 83/106). Sixty-seven records included no specific sampling depth. These records were generally more than 50 years old. Although they are likely near-surface sampling records and mainly report known shallow-water species, they cannot be verified as such because of the lack of explicit information. Most of these (87%, or 58/67) are records of scyphomedusae. Overall, the diversity of scyphomedusae was low, with only 39 species from 27 genera represented in records (Fig. 2A). The genus Chrysaora had the largest contingent of accounts, with 21 individual records of associations across at least seven Chrysaora species. This genus has been reported to interact with 16 different epifaunal crustaceans. The genera Chrysaora, Lychnorhiza, and Catostylus accounted for a third of scyphozoan records. These records originate mainly from the upper water levels of various locations (i.e., the east coast of the United States, the southeast of Brazil, the southern Australian coast, and the western Philippines, Japan and Pakistan).

Figure 2 Diversity of Scyphozoa and Hydrozoa species.

Rings from innermost to outermost are order, family, genus in the classes (A) Scyphozoa and (B) Hydrozoa as distributed by number of accounts including a host in that group. Families and genera with single reports are whitened.

Diversity of hydrozoan hosts

Twenty-six genera, and six Hydrozoan orders were reported interacting with Crustacea in 63 records (Fig. 2B). The order Leptothecata included the greatest number of records (18), with 17 records of Siphonophorae and 12 of Narcomedusae. The diversity of Hydrozoa was significantly limited by region, with 45 of the 63 records (71%) from the Gulf of California. Additionally, those from the Gulf were acquired from primarily deep water ROV missions. The medusae recorded belonged to 28 known species, with twelve records unable to provide higher resolution than genus and a single Prayid siphonophore only identified to the family level. Rosecea cymbiformis (Delle Chiaje, 1830) (4), Aegina citrea (Eschscholtz, 1829) (5), and Aequorea coerulescens (Brandt, 1835) (6) were the three most common species.

Diversity of crustacean epibionts

The crustaceans included Hexanauplia (reported in 37 discrete observations), Malacostraca (173), and a single representative of Branchiopoda (Evadne sp.) (Fig. 3). Recorded Hexanauplia consisted of mainly specialist groups known to be obligate epibionts and had overall low species resolution, with 13 of the 23 documented associations lacking a species name. The Macrochironidae, a group of known scyphozoan parasites, makes up 12 of the copepod epibiont records. Outside of this family, no additional Hexanauplia epibiont was recorded more than twice. The single reported case of a medusa with Evadne sp. occurred in a broad analysis of items found on a Catostylus medusae (Browne & Kingsford, 2005). As this was not replicated throughout medusae within the study, or in other studies, it is unlikely this is a common or genuine association.

Figure 3 Diversity of Crustacean epibionts.

From innermost ring to outermost ring: Subphylum, Order, Family, Genus. Color coded by classes Malacostraca (orange), Hexanauplia (pink), and Brachipoda (green). Families and genera reported only once are whitened.

The bulk of the associations involve crustaceans of the class Malacostraca. These 173 records include amphipods and decapods in equal proportion (47%, or 81/173 each), isopods (5%, or 9/173), and mysids (1%, or 2/173). The amphipods are dominated by the parasitic family Hyperidae, recorded in 32 separate encounters. Members of the family of Hyperidae are present across 22 identified scyphozoan and hydrozoan species, making them the most widely distributed family. Hyperia galba (Montagu, 1813) is present in nine records from both surface and deep-water samples, making it the single most plentiful within the amphipods. Outside of the family Hyperidae, Tryphana malmii (Boeck, 1871) is recorded six times in association with deep-sea jellyfish. Most amphipod species recorded were recorded on multiple host species.

Decapod associations (81 records) are separated among twelve families, Epialidae (17), Portunidae (14), Palaemonidae (12), Hippolytidae (14), Scyllaridae (11) Cancridae (6), Chlorotocellidae (2), Scyllaridae (1), Luciferidae (1), Penaeidae (1), Varunidae (1), and Grapsoidea (1). No decapod was found in association with hydrozoans or in deep-sea records. The representatives of Epialtidae are comprised exclusively of multiple species of the genus Libinia. The Portunidae records are mainly composed of the commercially valuable Charybdis feriata (Linnaeus, 1758) (11 records), Charybdis annulata (Fabricius, 1798) (1) and two Callinectes, Calinectes sapidus (Rathbun, 1896) and an unidentified Callinectes specimen (1). Periclimenes paivai (Chace, 1969) is the most common Palaemonidae, representing three of the twelve records, with six additional Periclimenes species, two Ancylomenes species and one Leander paulensis (Ortmann, 1897). All Hippolytidae associations were between a specimen of Latreutes anoplonyx (Kemp, 1914) or Latreutes mucronatus (Stimpson, 1860) and one of an array of different scyphomedusae in Asia, Australia, and the Arabian Sea-Persian Gulf corridor. The families Scyllaridae and Scyllarinae include seven Ibacus, three Scyllarus, and Eduarctus martensii (Pfeffer, 1881). These associations were all exclusively larval. The majority (4) of Cancridae records involve Metacarcinus gracilis (Dana, 1952) with two unknown Cancer species. These crabs were found on Chrysaora medusae and one Phacellophora camtschatica (Brandt, 1835). Two Chlorotocella gracilis (Balss, 1914) (Chlorotocellidae) were found on Japanese rhizostomes, both in somewhat limited encounters. The last three accounts include a Cyrtograpsus affinis (Dana, 1851) (Family: Varunidae), Lucifer sp. (Family: Luciferidae), and a juvenile Grapsoidea of unknown genus and species. The account of Lucifer sp. was of a record of one specimen on a medusa in New South Wales, and is not likely a common or genuine association (Browne & Kingsford, 2005). Cyrtograpsus affinis and the juvenile of the family Grapsoidea were also one-off reports found in single medusae (Schiariti et al., 2012; Gonçalves et al., 2016).

Associations that involved mysids or isopods were far fewer than those involving decapods and amphipods. The isopod records include only four species, including the deep-sea parasite Anuropus associated with Deepstaria enigmatica (Russell, 1967). Besides the in situ accounts of the Deepstaria scyphomedusae with an attached Anuropus, three Isopoda species were found in association with upper water column medusae. These are Cymodoce gaimardii (H. Milne Edwards, 1840) and Synidotea marplatensis (Giambiagi, 1922), each recorded three times, and Cymothoa catarinensis (Thatcher et al., 2003), found once in association with Chrysaora lactea (Eschscholtz, 1829). Within the order Mysida, the two species Mysidopsis cathengelae (Gleye, 1982) and Metamysidopsis elongata (Holmes, 1900) were recorded on Chrysaora during a bloom in the Southern California Bight (Martin & Kuck, 1991).

Three species of cirripeds were recorded 15 times in association with jellyfish, Alepas pacifica (Pilsbry, 1907) accounting for twelve of such records, Conchoderma virgatum (Spengler, 1789) accounting for two, and a single report of an unidentified Anelasma epibiont on a Pelagia noctiluca (Forsskål, 1775) from 1902. Alepas pacifica has been found on seven separate host species, all scyphozoans. The vast majority of these records came from a single literature review included within an extensive paper from Vader (1972). None of these species were found in deep-sea records.

Field collections

Only 58 papers included some explicit method of capture of the jellyfish and its epibiont (Fig. 4). Between 1862 and 1962, only seven of the twenty records reported a method of capture. From 1963 to 1989, this increased to 64%, with 25 of 39 records including the collection method. Since 1990, there have been only seven failures to report collection methods out of 140 accounts. The most common method of collection, used in 31 of the papers, is “by hand”, defined as using handheld dip nets, buckets, plastic bags, and, in limited cases, collection of carcasses from beaches. Trawling was first used in 1968 and has remained in use until recently, reported in 17 of the 33 associations after 2010. Although 38 records were obtained through deep water methods (HOV and ROV), these were used scarcely before 1999. Some studies employed multiple methods, with divers and ROV, or dip net and trawl capture, such that it was unclear which associations were found by each collection method. These were listed as “multi-method” and include four papers.

Figure 4 Collections information for both number of papers using a collection method and number of associations reported from this collection type.

Types are blue water diving (BWD), collection by hand (HC), multiple methods (MULTI), ring net (RN), scuba diving (SC), trawling (TR), in situ observation (OBS) or unknown (Unknown). Associations from papers in which multiple methods were used, but specific methods are known for each association are categorized under the known method. Many papers are comprised of multiple associations, as such, the “Individual” columns include each association separately, “Paper” columns report by paper.

The larger proportion of scyphozoan hosts to hydrozoan hosts may be a sampling artifact. The vast majority of the papers discussed here were only analyzing interactions in the top 30 m of the water column. A fair number, especially earlier texts, involve serendipitous encounters at the water’s edge or within sight of the surface (Bowman, Meyers & Hicks, 1963; Jachowski, 1963; Vader, 1972; Martin & Kuck, 1991). The larger, more visible nature of surface water scyphozoans of the rhizostomes and semaeostomes makes them an easier collection target than deep water species. Note that only a single scyphozoan of the order Coronatae, which has no large shallow representatives, was recorded as well. Many elements of the sampling methods impact the scope of this data, and the preeminence of hand collection and papers written on chance occurrences, as opposed to prolonged study, result in a picture that heavily weights organisms more frequently seen or interacted with by humans.

The oldest records of jellyfish-crustacean interaction involved hand collection with buckets and nets, often from shore. These include first accounts of hyperiid amphipod-jellyfish associations from the Chesapeake Bay (Bowman, Meyers & Hicks, 1963). Buckets and nets have remained mainstays, with hand collection accounting for 34 of the 108 post-2000 records and 32 of the 55 pre-2000 records. Buckets and plastic bags are likely preferable to nets, as they may reduce chances of epibiont detachment and medusa damage.

Trawling (by ring nets, otter nets, and bottom trawls), while reported in twelve papers, has been a prominent capture method in South America for the last two decades. However, trawling provides an additional threat, as epibionts may detach, get caught in the bell of a medusa, or move to a different location within the carcass. Given the damage sustained by gelatinous bodies during trawls, and the inability to capture more delicate associations, this is the methodology that seems most likely to provide low-quality relationship information. A focus on a lower number of medusae examined in more detail, may provide more useful information on the ecology of the interaction between jellyfish and their epibionts. Notably, Greer et al. (2017) uses a combination of in situ imaging (with an automatic ISIIS imaging system) and trawls. Trawls were used to verify the identity of organisms seen in the captured images. Such a protocol should be considered for future quantitative and qualitative work.

A total of 66% of the records (136/211) are from known surface encounters. 18% of the records (38/211) involve deep water accounts using either an ROV/HOV. These records are distributed unevenly across depths with few records below the mesopelagic zone (Fig. 5). Most of these records fail to provide epibiont location on the jellyfish but provide the only available information on deep water scyphomedusa and hydromedusa hosts. Most of the deep water records are from the Gulf of California. While this sampling method is useful, the high cost and difficulty of use of ROV and HOV equipment make it unrealistic for the vast majority of researchers. The limited number of deep-water accounts and the novelty of many of the findings on each dive can be attributed mainly to these limitations (Gasca & Haddock, 2004; Gasca, Suárez-Morales & Haddock, 2007; Gasca, Hoover & Haddock, 2015).

Figure 5 Percent of sampling by depth.

The depths of samples with known depths. 68% of samplings had known depth data (pie chart). 74.4% of sampling was done above 30 m. Where depth ranges were given (i.e., 8 to 30 m) the deeper value was used.

Given the fragility of scyphozoan and hydrozoan medusae, as well as the delicacy of the interaction with their epibionts, the most precise picture of the jellyfish-crustacean associations has been achieved from dip net, plastic bag, bucket, or other by-hand collection methods. These are not only a cost-effective strategy requiring little additional equipment, they also maintain maximum integrity of the organisms. Hand collection, however, is restricted to analyzing associations that are close to the surface. Trawl sampling provides a reliable way to collect many medusae offshore but sacrifices sample integrity. ROV is an imperfect sampling method, often failing to record epibiont positioning, but allows for the only viewing, documentation, and collection of deep water associations, thereby being uniquely important, especially for hydromedusa research. Moreover, the majority of the records document all symbionts on the target host species, often with little data beyond a name or tentative classification for the epibiont. This lack of closer examination leads to an inability to correctly categorize the nature of the relationship, including positioning, feeding behaviors, and duration of the interaction.

In conclusion, the overall best sampling results come from observation-first methodologies such as collection by-hand while snorkeling and diving, as in Mazda et al. (2019), ROV/HOV in situ underwater photography, as employed by Gasca, Hoover & Haddock (2015), or imaging and supplemental trawling as in Greer et al. (2017). Obtaining underwater pictures of medusae and epibiont is crucial to the understanding of the associate placement in relation to host and its behavior. It is also more informative than post hoc in-lab examinations and analysis of trawl contents, because the stress of collection and sampling may impact the epibiont position within the host (Hayashi, Sakagami & Toyoda, 2004). As waterproof video equipment becomes less expensive, options like a simple GoPro may provide clear enough imaging to allow novel in situ observations. Adding an underwater imaging component to sampling may also enable collectors to revisit the ecological context of the association.

Life stages

Age classes and sex, where available, are reported in Table 1. 63% of all records (133/211) reported an age class for the crustacean. 65% of the interactions with a listed age class (65%, or 86/133) reported crustacean juveniles, eggs, larval stages, copepodites, megalopae, or other immature forms. For a minority of records (37%, or 73/211), no information on the crustaceans’ age class and sex was available. When individuals were described as “male” or “female” without any qualifier attached, they were catalogued and treated as adult specimens (Table 1). Megalopae were noted only nine times out of the 106 records that reported an age class for the crustacean associate (8%). In these nine records, the megalopae belonged to the genera Callinectes, Periclimenes, Metacarcinus, Cancer, and Charybdis, and were all in association with Scyphomedusae (Orders: Rhizostomeae and Semaeostomeae). In addition to megalopae, phyllosoma larvae of the families Scyllaridae and Scyllarinae were reported 12 times. The occurrence of larvae of this type associated with medusae and, more generally, with gelatinous zooplankton is well known, especially along the Japanese coast (Wakabayashi, Tanaka & Phillips, 2019). Within and upon the host, juvenile crustaceans were often coexisting with adult forms. Eighty-one of the associations include juveniles (excluding megalopae, eggs, and copepodites), sometimes embedded in host tissue (Towanda & Thuesen, 2006; Browne, 2015; Yusa et al., 2015; Browne, Pitt & Norman, 2017; Mazda et al., 2019). The presence of eggs and ovigerous females was reported in 39 cases from 23 different species. In at least three papers, females and ovigerous females were present in exceptionally high proportions relative to adult males (Filho et al., 2008; Oliva, Maffet & Laudien, 2010; Mazda et al., 2019). Records of megalopae of the commercial crab, Charybdis feriata were reported in substantial numbers on two separate hosts (Kondo et al., 2014; Boco & Metillo, 2018). In other reports, associations between juvenile Metacarcinus gracilis (Dana, 1852) and medusae are hypothesized to be beneficial to the crab as the medusae supply means of transport and food acquisition, which may be similar across juvenile decapod-scyphozoan associations (Towanda & Thuesen, 2006).

Nature of associations between medusae and crustaceans

There is no agreement between authors on the degree to which medusae and crustaceans’ interactions are parasitic, commensal, or otherwise. In the case of the scyphozoan Phacellophora camtschatica and the decapod Metacarcinus gracilis (Dana, 1852), the interaction may involve a mutualistic cleaning relationship as M. gracilis graduates into adulthood (Towanda & Thuesen, 2006). Other reports of megolopae do not suggest any parasitization of the medusae. Weymouth (1910) also indicates that this is a commensal relationship important to M. gracilis megalopae until they reach ~20mm. In other cases, such as the shrimp Perimincles paivai, the commensals seemed to be feeding on the mucus, not the host tissue (Browne & Kingsford, 2005; Filho et al., 2008). Dittrich (1988) demonstrates an aggressive parasitoidism by Hyperia galba in which a large subset of host medusae was so reduced by predation as to lose almost all morphological features. While the ultimate death of these hosts is not recorded within the text, the loss of all tentacular structure and non-mesoglear tissue would make survival nearly impossible. The numbers in which Hyperia can be found on some of the recorded medusae, occasionally upwards of 100 amphipods engaging in host consumption, may lend credence to the parasitoid rather than classically parasitic nature of this relationship in many hosts (Vader, 1972; Dittrich, 1988; Towanda & Thuesen, 2006). However, additional reports on the same species and other hyperiids reported that this group engages in cradle positioning, facing outwards from the medusa, into the water column with no reported predation, or engage in only limited predation of the gonadal tissue or mesogleal tissue (Bowman, Meyers & Hicks, 1963; Gasca 2005; Browne, 2015). Based on this information it seems likely that the family Hyperidae includes a variety of strategies, and the family Hyperia itself may also encompass non-aggressive parasitism, aggressive parasitism, and parasitoidism. In part, this may be due to temporal behavioral differences within species, with more extreme predation in summer and autumn and limited parasitism in spring as populations raise and fall (Bowman, Meyers & Hicks, 1963; Dittrich, 1988). “Inverted cradle” positioning is a recurring feature of amphipod associates (Bowman, Meyers & Hicks, 1963; Condon & Norman, 1999). While some of the crustaceans fed on the medusae themselves, Towanda & Thuesen (2006) primarily recorded crustaceans engaging in theft of prey collected by medusae. Many crustaceans that were reported feeding on the medusae were feeding entirely or in part on the highly regenerative gonadal tissue (Pagès, 2000; Towanda & Thuesen, 2006; Ohtsuka et al., 2009) or engaging in the excavation of small pits in the host mesoglea (Humes, 1953; Jachowski, 1963; Browne, 2015). Reports of Libinia dubia (H. Milne Edwards, 1834) have the greatest agreement on the parasitic nature of the species’ interactions with their medusa host (Jachowski, 1963; Phillips, Burke & Keener, 1969; Schiariti et al., 2012).

The largest exception to the above patterns of limited consumption or longer term residence is the scholarship surrounding phyllosoma larvae on gelatinous zooplankton. These larvae have been reported to stab a pair of pereiopods through the exumbrella or exterior of a nectophore and use the medusa as propulsion and food source. This is a common occurrence both in the northern Gulf of Mexico and at various locations along the Japanese coast (Greer et al., 2017; Wakabayashi, Tanaka & Phillips, 2019). In the review on the subject by Wakabayashi, Tanaka & Phillips (2019), it is hypothesized that the flattened body and ventral mouth of these phyllosoma larvae is ideal for consumption of gelatinous zooplankton while attached. The exact length of this parasitoid association is unknown, though it is likely generally ended by the medusa’s eventual death as the larva eats its way through.

The degree to which crustaceans engage in host consumption may be in part obscured by the speed with which medusae regenerate tissues, especially gonadal and oral arm tissues (Towanda & Thuesen, 2006). The number of associates (at least eight crustacean species) found residing within the bell and around the gonads, suggests that gonadal tissue may be common nourishment even when bell and arm tissue is not consumed. Overall, the relationships of crustaceans with their medusa hosts remain largely uncharacterized and require additional study. Few papers have analyzed the gut contents of the epibionts, which would be a helpful tool in determining whether inverted positioning on hosts was actually a signal of lack of consumption, or simply a break from such (Vader, 1972; Pagès, 2000; Towanda & Thuesen, 2006; Oliva, Maffet & Laudien, 2010). Detailed records of the diets of such organisms are difficult to reconstruct. However, specific searches for nematocysts in digestive tract and excretions or stable isotope analysis have proven successful at identifying cnidomedusae as possible food sources (Schiariti et al., 2012; Fleming et al., 2014). Expanding future works to include both these practices, photographs of the host medusae, and notes on swimming strength, tentacular loss and other signs of deterioration would improve our understanding of how detrimental these relationships actually are. This sort of documentation of host condition is impossible when specimens are collected via trawl.

In addition to consumption, the issue of host choice and host specificity has been analyzed only sparsely. There is evidence in multiple studies that while some individual jellyfish host symbionts, others in the same area lack them due to their size or species (Towanda & Thuesen, 2006; Ohtsuka et al., 2011; Boco & Metillo, 2018). While exotic species often have lower amounts of parasitization in their introduced range (Torchin et al., 2003), the degree to which epibionts in medusae are affected by host or epibiont endemicity is unknown. The high number of cryptic species, a history of misidentification, and poor understandings of historical ranges compound issues with sparse research on the topic (Dawson, 2005; Graham & Bayha, 2007; Morandini et al., 2017; De Souza & Dawson, 2018).

Only one study provides an indication of how nuanced the relationship between gelatinous zooplankton hosts and epibionts may be; 6 years of monthly observation showed that single adult females of the amphipod Oxycephallus clausi (Bovallius, 1887) had a broad range of gelatinous hosts, but shifted to primarily Ocyropsis fusca (Rang, 1827), a lobate ctenophore, during brood release (Mazda et al., 2019). While ctenophores are not the focus of this review, it shows that the nature of interactions may change during the crustacean lifecycle. These sorts of long-term analyses are hard to pursue, but provide a fascinating look at the range of information that can be collected with observational methods. Uneven sex ratios, such as those seen in the case of Oxycephallusclausi (97% female), are present across many associations (Condon & Norman, 1999; Filho et al., 2008; Oliva, Maffet & Laudien, 2010; Mazda et al., 2019). The most common explanation for this higher ratio of females and often ovigerous females is use of scyphozoan and hydrozoan hosts primarily as nursery habitat for movement and protection of juveniles (Gonçalves et al., 2016; Gonçalves et al., 2017; Mazda et al., 2019). Potential territoriality in some females, like those of P. paivai, may help ensure more resources for their brood, and is in line with other symbiont crustaceans (Baeza et al., 2017). For deep sea crustaceans, such as Pseudolubbockia dilatata (Sars, 1909), more even sex ratios would be expected, as there is evidence of long-term resident brooding pairs, and mate scarcity is a feature of deep sea life. Evidence for long-term association and pairing has not been found for other deep water crustaceans, although understanding these deep sea interactions is generally hampered but small sample sizes and difficulty of observation (Gasca, Suárez-Morales & Haddock, 2007; Baeza et al., 2017; Gasca & Browne, 2018).

Years and locations

The oldest records examined were only available from earlier literature reviews (Pagès, 2000; Towanda & Thuesen, 2006; Schiariti et al., 2012). The first record is the Bate (1862) account of the amphipod Iphimedia eblanae on the scyphozoan Rhizostoma pulmo (Macri, 1778) from 1862, also reported in the Vader (1972) review on amphipod associations with medusae. Thiel (1976) refers to older records from as far back as 1791. Overall, the number of records detailing interactions has risen over time but has not exceeded ten papers during any 5 years. While these numbers are increasing modestly, the number of distinct interactions that any given paper reports have increased. Pre-1990s articles, on average put forward information on 1.24 associations per paper. In contrast, the average number of associations reported in papers published from 1990 to 2018 increased more than twofold (an average of 2.83 records per paper). These surveys provide useful records of separate associations found in one area or on one organism and are informative of ecosystem features on a regional level. Still, given the studies’ breadth, they often lack depth, not characterizing relationships between individual host species and their associates.

Records were unevenly distributed globally, with Africa and Europe completely devoid of records from the past 30 years with the exception of a single note on an accidental observation from Gran Canaria, Spain. The eastern coast of North America (one record since 1984 (Tunberg & Reed, 2004) and China (no direct records)), as well as West Africa (one record from 1972 (Bruce, 1972)) and the Mediterranean Sea (last collections 1985 (Dittrich, 1988)) also lack records from the last 30 years. The areas consistently covered by recent papers are Australia (1968–2009), the Philippines (2014, 2018), the eastern coast of South America (1980–2016), and the western United States (1966–2015). Japanese records represent the longest continuity over time, with 33 records between 1902 and 2019. The association that consistently appears throughout time is that of Alepas pacifica (Thoracica, Lepadiformes) with Nomura’s Jellyfish (Nemopilema nomurai) (Pagès, 2000; Yusa et al., 2015). The first record of this association was in 1902 (Pagès, 2000), and the most recent in 2015 (Yusa et al., 2015). Phyllosoma larvae of multiple species, Chlorotocella gracilis (Balss, 1914), and Latreutes spp. also have records spanning multiple decades and papers.

It is worth mentioning that the uneven geographic distribution of associations reported herein may be an artifact of lack of readily available English translations of works from some areas. Reports from Japan and China of crustacean and gelatinous zooplankton associations are mentioned by Hayashi, Sakagami & Toyoda (2004) and Wakabayashi, Tanaka & Phillips (2019), but were not available in English and therefore are not accounted for in this review. Similarly, European records may be underestimated, as non-English records are absent. Other locations’ lack of records may be a more accurate representation of a gap in academic knowledge. Africa’s west and eastern coasts are known to be understudied ecosystems, and so the missing research here is likely not just untranslated (Berkström et al., 2019). As in other ecological inquiries, the expansion of Local Ecological Knowledge into the study of gelatinous zooplankton should be considered, as fishermen and coastal communities often have a deep knowledge of organisms and their associations (Berkström et al., 2019). Fishermen are often well acquainted with specific gelatinous zooplankton species and know their harms, and may have knowledge of symbionts living upon or within them (Al-Rubiay et al., 2009).

Commercial species

Many commercial crustaceans and jellyfish were found to have associations that may be of ecological and commercial importance. Twelve records reported the edible jellyfish Rhopilema spp. as hosts (Berggren, 1994; Pagès, 2000; Hayashi, Sakagami & Toyoda, 2004; Towanda & Thuesen, 2006; Ohtsuka et al., 2010; Ohtsuka, Boxshall & Srinui, 2012; Boco & Metillo, 2018). The commercially harvested shrimp, Penaeus stylirostris (Stimpson, 1871), was found on Stomolophus meleangris (Riascos et al., 2018). Notably, young Callinectes sapidus, the Chesapeake Blue Crab, was reported by Jachowski (1963) as regularly found on Chrysaora quinquecirrha (Desor, 1848) medusae without consuming them. This association was reported again briefly in the Mississippi Sound by Phillips, Burke & Keener (1969). This interaction between a jellyfish and the blue crab has never been corroborated further except for a nonspecific report of a Callinectes sp. associated with jellyfish reported by Towanda & Thuesen (2006) as unpublished data. The commercially valuable crab, Charybdis feriata, has been reported in association with ten jellyfish species (Berggren, 1994; Towanda & Thuesen, 2006; Ohtsuka et al., 2010; Schiariti et al., 2012; Boco, Metillo & Papa, 2014; Boco & Metillo, 2018). These reports involve juveniles (Trott, 1972; Towanda & Thuesen, 2006; Schiariti et al., 2012; Kondo et al., 2014; Boco & Metillo, 2018) and megalopae (Kondo et al., 2014; Boco & Metillo, 2018) of C. feriata, and this association has been recorded in Hong Kong, Japan, the Philippines, Mozambique, and Indonesia, suggesting a consistent pattern over time (first record in 1965 (Schiariti et al., 2012) and last record in 2014 (Boco & Metillo, 2018)) and across their range.

Slipper lobster larvae of the genera Scyllarus and Ibacus have been reported many times across various hosts (Wakabayashi, Tanaka & Phillips, 2019). Some slipper lobsters are commercially fished for consumption, and a large number of these larvae (40% in the Gulf of Mexico) have been shown to live attached to gelatinous zooplankton (Greer et al., 2017).

The consumption of some Scyphozoan hosts, such as Catostylus mosaicus and Rhopilema spp., makes their records valuable as well. The fishing pressures on the jellyfish populations may significantly impact the crustaceans that rely on their oral arms and bells for transport and nourishment of their juvenile stages. Further understanding of these relationships may be especially important in cases where both the medusae (e.g., Rhopilema spp., Lobonemoides robustus (Stiasny, 1920) and Catostylus spp.) and crustacean (Charybdis feriata) are subject to fishing (Boco, Metillo & Papa, 2014; Boco & Metillo, 2018, Kondo et al., 2014). Finally, current information on Callinectes sapidus and its relationship to and frequency of interaction with host jellyfish is needed, as the blue crab represents a commercially valuable fishery in the Gulf of Mexico and along the Atlantic Coast of the USA.

Understanding the nature of the relationships between economically valuable species of Crustacea and common scyphozoans and hydrozoans can improve fisheries practices and regulation, as already acknowledged for economically important fish and their jellyfish hosts (Tilves et al., 2018). The importance of maintaining juvenile communities for commercially sized adult populations to recruit from is well established and a frequent impetus for marine protection areas. The fishing of medusae is different from most modern vertebrate fishing. It is temporally highly variable, and blooms, when found, are fished as intensely as possible by local fishermen. It is also comparatively new as an export industry, especially in Southeast Asia (Omori & Nakano, 2001). Additional regulation and management should be considered for jellyfish species known to harbor juveniles of commercially viable crustaceans. It is clear that many crustaceans, fish, and other organisms live in, upon and around medusae, thus indiscriminate efforts to remove or destroy blooms of endemic species are likely unwise (Tilves et al., 2018; Riascos et al., 2018).

Conclusion

Many of the interactions we reviewed are fragmented and not comprehensive. Studies covering timing and breadth of infection of commercially valuable crustaceans on marine scyphozoans are scarce, but may be valuable information to fully understand the complexity of their life cycle, and thus the species’ vulnerability at each life cycle stage. The general picture of the commensal relationships that arise from this review is complex and emphasizes the diversity of jellyfish and crustaceans’ relationships. Any attempt to paint them as uniformly parasitic fails to acknowledge the diversity of crustacean host-use strategies. While some seem to be parasitic or parasitoid, others are life-stage dependent commensals reliant on medusae for transportation. Some deep water crustaceans may be lifelong commensals (Gasca, Suárez-Morales & Haddock, 2007). In each of these cases, the work thus far is far from exhaustive. Additional research on seasonality, maternal care, territoriality, impact on host and other such matters should be further pursued.

The scyphozoans and hydrozoans studied here represent only a small proportion of the globally recognized species. Even shallow water coastal species are poorly covered. This research has been restricted to a small selection of near-shore sites over the past 50 years, leaving inadequate coverage even in regions with a significant scyphozoan research presence (i.e., the Mediterranean, western Europe, China, northeastern North America). Because much of the published research focused on single occurrences, this paper’s overall results do not necessarily capture the broader ecology of the species involved (Bowman, Meyers & Hicks, 1963; Jachowski, 1963; Suzuki, 1965; Ohtsuka et al., 2011). Similarly, species descriptions that mention an association without details on the conditions in which it was found offer little insight on the frequency and ecological role of such interactions (Humes, 1953; Reddiah, 1968; Bruce, 1972; Criales, 1984; Bruce, 1988; Bruce, 1995; Bruce, 2008).

Best practices moving forward should include some of the following elements: in situ imaging pre-collection, observations on medusa health, analysis of epibiont gut contents when possible, preferential use of non-destructive collection methods, observations on symbiont placement within or upon the medusa, and frequency, geographical and temporal variation of the association.

With this review, we hope to highlight a significant knowledge gap and a lack of formal study on the ecology of the crustaceans residing on and around jellyfish, as well as a glimpse of the ecological complexity of these interactions. We provide easy access to a century of ecological research and a framework for analyzing and contextualizing future research on this topic.

Supplemental Information

Supplemental Information 1 Expanded medusa crustacean association table.

Every association in all reviewed papers with details on species and higher order classification of host, species of associate, sex and life stage of associate, notes on association, location on host, location association was recorded, date of record, depth of association and literature source. Expanded to include higher taxon labels for both crustaceans and medusae.

Click here for additional data file.

Additional Information and Declarations

Competing Interests

Author Contributions

Data Availability

The authors declare that they have no competing interests.

Kaden Muffett conceived and designed the experiments, performed the experiments, analyzed the data, prepared figures and/or tables, authored or reviewed drafts of the paper, and approved the final draft.

Maria Pia Miglietta conceived and designed the experiments, authored or reviewed drafts of the paper, and approved the final draft.

The following information was supplied regarding data availability:

The raw data is available in the Supplementary Table and the primary tables in the article.

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
