# Peer review of "Planktonic associations between medusae (classes Scyphozoa and Hydrozoa) and epifaunal crustaceans"

_PeerJ, doi:10.7717/peerj.11281_

## Round 0.1 · original submission · Major Revisions

This is an important revision paper that summarizes the information available on the relationships between medusa and crustaceans, however there are some points that need to be addressed and the work would also benefit from a more in-depth analysis of the published information used. The present version of the review seems a simple description of the information extracted on the papers gathered from the search. Some discussion and critical analysis of the results are necessary. Therefore, I agree with Reviewer 3 that comments and discussion of the results are necessary. Some points are, e.g., why there are no studies and/or reports on the associations between medusa and crustaceans for certain geographical areas, as European waters? a discussion on the sampling methods used so far to collect information on the subject and, on the different types of associations between crustaceans and jellyfish. As Reviewer 1 points out, there are some associations that need to be discussed and categorized, e.g. the Evadne sp. and Lucifer sp.. This same reviewer highlight some references that were not used in the review, that I recommend to be added to the analysis.

Reviewer 1 ·

Basic reporting

This study totally depends on access to Google Scholar. Unfortunately, your actions were not enough. Use other key words to find more important papers. Can you find any interaction specific to crustaceans?
Many other invertebrates and fish are associated with jellyfish. They seem to utilize jellyfish for antipredation, dispersal, food and so on. For example, fish juveniles seem to mainly use the hosts for protection and food.

Experimental design

The aims and scope of this paper are clear. Although you totally depend on Google Scholar, your trials were not sufficient. Use other key words to find more informative and important papers.

Validity of the findings

You have succeeded in finding the generalized patterns in interactions between jellyfish and crustaceans. However, more and more exact citations are definitely needed. Some important papers are missing.

Additional comments

This is an overview dealing with interactions between jellyfish and epibiont crustaceans. It is very informative for future works. However, your access to papers is not sufficient. Many important papers are missing (see attached file) and citation is not enough. You must read cited papers more carefully. In addition, presumed interactions such as parasitoidism are suggested for these relationships. Summarize these, too.

Annotated reviews are not available for download in order to protect the identity of reviewers who chose to remain anonymous.

Reviewer 2 ·

Basic reporting

It has some minor deficiencies in the literature cited, as written below:
271 typing mistake, it must be "Pagès"
272-274 confusing bibliographic reference
274-275 duplicated bibliographic reference
299-300 lacking bibliographic reference
299 typing mistake, it must be "stylirostris"
301-302 duplicated bibliographic reference
303 duplicated bibliographic reference
305-306 duplicated bibliographic reference
548 No alphabetical order
An some other anotations

Experimental design

No coment

Validity of the findings

no coment

Additional comments

This manuscript is an excellent compilation of the state of the art in relation to associations between invertebrates and, due to how difficult it is to publish about this type of interactions, (to find them you depend a lot on luck!) There is no doubt that it will become in a reference work for future publications, both to make comparisons or as a source of specialized bibliography.

Annotated reviews are not available for download in order to protect the identity of reviewers who chose to remain anonymous.

Reviewer 3 ·

Basic reporting

A clear and professional English is used throughout the manuscript; the references and background are sufficient to introduce the subject of this work.

I consider it pertinent to discuss some aspects that I detail below; Finally, figures are relevant to the content of the work, but please improve the quality of figure 4, if possible, as well.

I noticed that at least the reference Tapia et al. 2018 has an error. The correct form is Puente-Tapia et al. 2018. Modify both in the references section, as well as in the table (check all the references if appropriate):

Puente-Tapia FA, Gasca R, Genzano G, Schiariti A, Morandini AC. 2018. New records of association between Brachyscelus cf. rapacoides (Arthropoda: Amphipoda) and medusae (Cnidaria: Scyphozoa and Hydrozoa) from São Sebastião Channel, southeast Brazil Francisco. Brazilian Journal of Oceanography 66:301–306....

please delete Francisco in the reference

Experimental design

Congratulations to the authors. This manuscript represents an updated review of the knowledge about the several types of biological associations between medusae and epifaunal crustaceans worldwide.

The present work described with sufficient detail the methods and information to replicate. This work made an exhaustive search of available information with a detailed “discrimination” of the sources to be considered in the review, which allows the provided information to be of high quality. In general, the manuscript well written easy to understand the methodology, results, and discussions. The different subsections of the results-discussion allow a clear understanding of each of the aspects that the authors intend to highlight.

I only have a suggestion on the methodology section: Although the main objective of this work is not a taxonomic aspect, I consider important to mention the taxonomic proposal used to indicate the taxonomic classification of each species (e.g., WoRMS, among others), because in some species, such as the medusa Liriope tetraphylla, depending on the source it is classified as Limnomedusae or Trachymedusae.

Validity of the findings

Conclusion are well stated, however, I think this section could be expanded with some suggestions that I provide below.

Additional comments

Dear Editor:
Please accept this review of the manuscript entitled: “Planktonic association between medusae (classes Scyphozoa and Hydrozoa) and epifaunal crustaceans” by Kaden McKenzie Muffett and María Pia Miglietta.

Sometimes the reporting of biological interactions is due to fortuitous findings of the organisms involved in the interactions, and not due to the main objective of the research (i.e., in studies where the spatiotemporal variation of the medusae community is described); therefore, the methodologies employed are not always the most adequate for the detailed description of the biological association. Please mention if it is possible to conclude if this factor is determinant so that many of the interactions cannot be described in detail, i.e., the total number of individuals of the crustaceans, aspects such as prevalence and intensity of infection values, among others, as well as the location of the crustacean in the host, the possible interaction with other groups of organisms, etc. The fact that a spatiotemporal discrepancy is observed in the study of the interaction between medusae and crustaceans, can be an indicator that these works are the product of the fortuitous findings between both groups.

The authors mention that Scyphomedusae species are mentioned more frequently than Hydromedusae in association with crustaceans, so please mention if the type of sampling (coastal, shallow waters, open sea, and deep waters) its feasibility, or the ecology of the host and crustacean are the determining factors for more reports of the scyphozoan compared to hydrozoan.

If it is possible, the authors could mention that the description of the biological association between species with commercial importance represents an essential tool to understand the ecology-biology of these species, which can help in the generation of fishing regulation (in case of species of import, export or local consumption), as the edible medusae Stomolophus meleagris or Lychnorhiza lucerna, as well as some shrimps or crabs.

I agree with the authors that possibly the best methodologies of collection to document the different types of biological associations between medusae and crustaceans are the dip net, plastic bag, bucket, or by-hand collection methods, even though these methodologies are limited to superficial observations. Although each type of collection methodology has certain advantages and disadvantages, I believe that it would be appropriate for the authors to expand the discussion on the advantages/disadvantages of the different methods, which could be helpful for future research on medusa specimens and their epibionts. For example, the ROVs and their videofilms allow to see the naturalness of the association and, in the case the medusa is collected, the specimen is captured without corporal damage, but it has the disadvantage of the high cost and only some institutes have access to this type of sampling tools.

As a complement to the above, the authors may conclude or suggest for future research, the possibility of analyzing the biological interaction in vivo (within the possibilities of each study) to try to prevent the epibiontic crustacean escaping or being lost by manipulation of the zooplankton sample or the medusa host.

Although a significant number of papers reported some of the effects of the crustacean on its medusa host, the authors could include a recommendation or suggestion that future works should analyze the possible harms or benefits to the host, or analyze the stomach contents of the crustaceans to determine if there is some degree of predation on the host or to the host’s prey, since the associations are categorized based on the detail of the analysis. As mentioned by the authors, another type of analysis is needed to define more accurately the type of damage or interactions between the two groups of organisms. In small crustaceans, such as amphipods, in which it is difficult to analyze the gut content, stable isotopes analysis of the stomach contents can be performed to determine whether they consume the host or the host´s prey.

Although the objective of this work is the revision of the different association between medusae and crustaceans, there are several aspects about the biology of each species that could be commented, such as the fact that females and larval-juvenile stages have been observed more frequently than males, which may be due to the female use the host as a refuge, transport and “nursery” for their offspring, while the males, as in the case of hyperids, leave the host to try to reproduce and feed.

---

## Round 0.2 · Minor Revisions

I am impressed with the revision made as the new version has improved. However, I sent the MS to reviewer 1 and he agrees on that. Therefore, I recommend a last revision to comply with the PeerJ guidelines to authors and to correct some minor mistakes. After that, I will be happy to accept this review paper for publication.

Reviewer 1 ·

Basic reporting

My impression is "well done", but there are many careless mistakes. References must be arranged in chronological order throughout the text (e.g., L344, 432-435, 486-487 and maybe others). A careless mistake is found on L319. MPA (L457) should not be abbreviated. In Fig. 4, what do you mean by "individual"? Explain it more explicitly.

Experimental design

Much more references are added to analyze relationships between jellyfish and crustaceans. Not only commercially harvested jellyfish but also non-indigenous species may be discussed. What happens to symbionts after introduction of hosts to new habitats (if data are available)? Anthropological impacts on symbioses may be discussed in more details.

Validity of the findings

Present problems and perspectives are well discussed.

Additional comments

If careless mistakes are amended, it will become acceptable sooner or later. Anyway, this is very informative for future studies on the marine ecosystems involving jellyfish. Again, take a look at the guideline for authors to improve the manuscript.

---

## Round 0.3 · accepted · Accept

I'm confident that this review will be a good help for anyone interested and dealing with the subject in future studies.